# STYLESHOT: A SNAPSHOT ON ANY STYLE

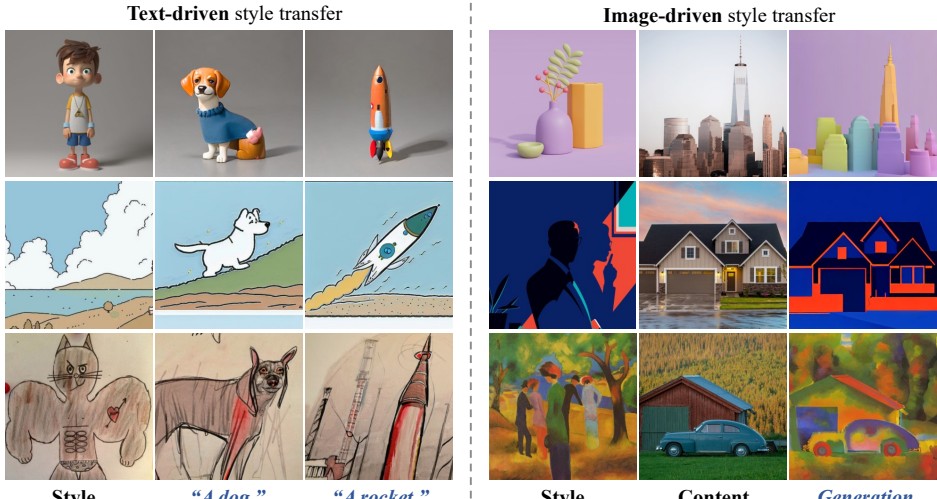

**Text-driven** style transfer        **Image-driven** style transfer

**Style**    *"A dog."*    *"A rocket."*     **Style**    **Content**    *Generation*

Figure 1: Visualization results of **StyleShot** for text and image-driven style transfer across six style reference images. Each stylized image is generated by StyleShot without test-time style-tuning, capturing numerous nuances such as colors, textures, illumination and layout.

## ABSTRACT

In this paper, we show that, a good style representation is crucial and sufficient for generalized style transfer without test-time tuning. We achieve this through constructing a style-aware encoder and a well-organized style dataset called Style-Gallery. With dedicated design for style learning, this style-aware encoder is trained to extract expressive style representation with decoupling training strategy, and StyleGallery enables the generalization ability. We further employ a content-fusion encoder to enhance image-driven style transfer. We highlight that, our approach, named StyleShot, is simple yet effective in mimicking various desired styles, i.e., 3D, flat, abstract or even fine-grained styles, *without* test-time tuning. Rigorous experiments validate that, StyleShot achieves superior performance across a wide range of styles compared to existing state-of-the-art methods.

## 1 INTRODUCTION

Image style transfer, extensively applied in everyday applications such as camera filters and artistic creation, aims to replicate the style of a reference image. Recently, with the significant advancements in text-to-image (T2I) generation based on diffusion models (Ho et al., 2020; Nichol & Dhariwal, 2021; Nichol et al., 2021; Ramesh et al., 2022; Rombach et al., 2022; Wang et al., 2024b), some style transfer techniques that build upon large T2I models show remarkable performance. Firstly, style-tuning methods (Everaert et al., 2023; Lu et al., 2023; Sohn et al., 2024; Ruiz et al., 2023; Gal et al., 2022; Zhang et al., 2023) primarily tune embeddings or model weights during test-time. Despite promising results, the cost of computation and storage makes it impractical in applications.

Even worse, tuning with a single image can easily lead to overfitting to the reference image. Another trend, test-time tuning-free methods (Fig. 2 (a)) (Wang et al., 2023b; Liu et al., 2023; Sun et al., 2023; Qi et al., 2024) typically exploit a CLIP (Radford et al., 2021) image encoder to extract visual features serving as style embeddings due to its generalization ability and compatibility with T2I models. However, since CLIP image encoder is primarily trained to extract unified semantic features

with intertwined content and style information, these approaches frequently result in *poor style representation*, with detailed experimental analysis in Sec. 4.4. Moreover, some methods (Liu et al., 2023; Ngweta et al., 2023; Qi et al., 2024) tend to decouple style features in the CLIP feature space, resulting in unstable style transfer performance.

To address the above limitations, we propose **StyleShot**, which is able to capture any open-domain styles without test-time style-tuning. First, we highlight that proper style extraction is the core for stylized generation. As mentioned, frozen CLIP image encoder is insufficient to fully represent the style of a reference image. A **style-aware encoder** (Fig. 2 (b)) is necessary to specifically extract more expressive and richer style embeddings from the reference image. Moreover, high-level styles such as 3D, flat, etc., are considered global features of images. It is difficult to infer the high-level image style from small local patches alone, which motivates us to extract style

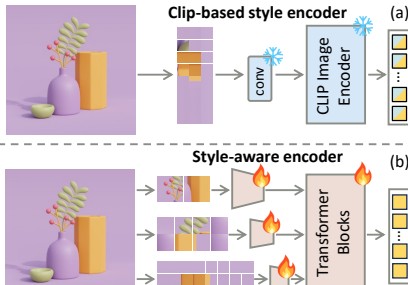

Figure 2: Illustration of style extraction between CLIP image encoder (a) and our style-aware encoder (b).

embeddings from larger image patches. Considering both low-level and high-level styles, our style-aware encoder adopts a Mixture-of-Expert (MoE) structure to extract multi-level patch embeddings through lightweight blocks for varied-size patches, as shown in Figure 2. All of these multi-level patch embeddings contribute to the expressive style representation learning through task fine-tuning. Furthermore, we introduce a novel **content-fusion encoder** for better style and content integration, to enhance StyleShot's capability to transfer styles to content images.

Second, a collection of style-rich samples is vital for training a generalized style-aware encoder, which has not been considered in previous works. Previous methods (Wang et al., 2023b; Liu et al., 2023) typically utilize datasets comprising predominantly real-world images (approximately 90%), making it challenging to learn expressive style representations. To address this issue, we have carefully curated a style-balanced dataset, called **StyleGallery**, with extensive diverse image styles drawn from publicly available datasets for training our StyleShot, as detailed in the experimental analysis in Sec. 4.4.

Moreover, to address the lack of a benchmark in reference-based stylized generation, we establish a style evaluation benchmark **StyleBench** containing 73 distinct styles across 490 reference images and undertake extensive experimental assessments of our model on this benchmark. These qualitative and quantitative evaluations demonstrate that StyleShot excels in transferring the detailed and complex styles to various contents from text and image input, showing the superiority to existing style transfer methods. Additionally, ablation studies indicate the effectiveness and superiority of our framework, offering valuable insights for the community. We further demonstrate the remarkable ability of StyleShot in learning fine-grained styles.

The contributions of our work are summarized as follows:

- We propose a generalized style transfer method StyleShot, capable of generating the high-quality stylized images that match the desired style from any reference image without test-time style-tuning.

- To the best of our knowledge, StyleShot is the first work to designate a style-aware encoder based on Stable Diffusion and a content-fusion encoder for better style and content integration.

- StyleShot highlights the significance of a well-organized training dataset with rich styles for style transfer methods, an aspect that has been overlooked in previous approaches.

- We construct a comprehensive style benchmark covering a variety of image styles and perform extensive evaluation, achieving the state-of-the-art text and image-driven style transfer performance compared to existing methods.

## 2 RELATED WORK

**Large T2I Generation.** Recent advancements in large T2I models have showcased remarkable abilities to produce high-quality images from textual inputs. Specifically, diffusion based T2I models

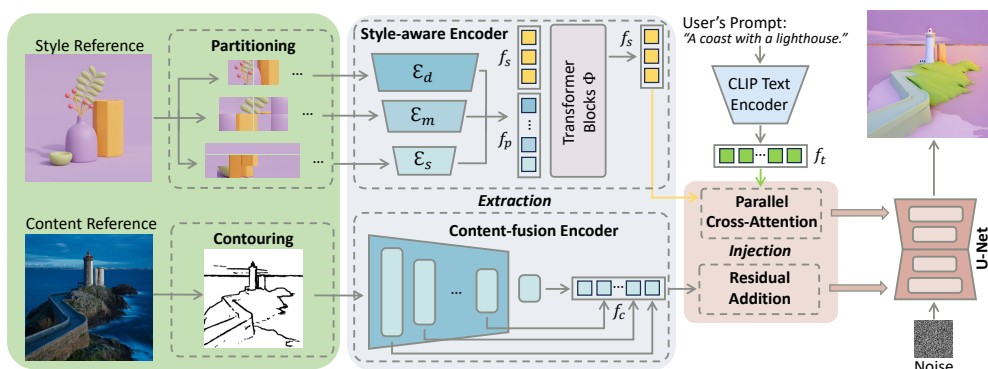

Figure 3: The overall architecture of our proposed StyleShot.

outperform GANs (Radford et al., 2015; Mirza & Osindero, 2014; Goodfellow et al., 2020) in terms of both fidelity and diversity. To incorporate text conditions into the Diffusion model, GLIDE (Nichol et al., 2021) first proposed the integration of text features into the model during the denoising process. DALL-E2 (Ramesh et al., 2022) trained a prior module to translate text features into the image space. Moreover, studies such as Ho & Salimans (2022) and Dhariwal & Nichol (2021); Go et al. (2023) introduced classifier-free guidance and classifier-guidance training strategies, respectively. Following this, Stable Diffusion (Rombach et al., 2022) utilizes classifier-free guidance to train the diffusion model in latent space, significantly improving T2I generation performance. Our study aims to advance stable and efficient style transfer techniques on the superior image generation capabilities of large diffusion-based T2I models.

**Image Style Transfer.** Image style transfer aims to produce images that mimic the style of reference images. With deep learning's evolution, Huang et al. (2018); Liu et al. (2017); Choi et al. (2018); Zhu et al. (2017) introduced unsupervised method on GANs (Heusel et al., 2017) or AutoEncoders (Hinton & Zemel, 1993; He et al., 2022) in explicit or implicit manner for automatic style domain conversion using unpaired data, ensuring content or style consistency. Furthermore, another research avenue (Gatys et al., 2016; Ulyanov et al., 2016; Dumoulin et al., 2016; Johnson et al., 2016) utilized the expertise of pre-trained CNN models to identify style features across different layers for style transfer. Nonetheless, the limitations in generative performance of conventional image generation models like GANs and AutoEncoders often result in subpar style transfer results.

Leveraging the exceptional capabilities of large T2I models in image generation, numerous style transfer methods have exhibited remarkable performance. Style-tuning methods (Everaert et al., 2023; Lu et al., 2023; Gal et al., 2022; Zhang et al., 2023; Ruiz et al., 2023; Sohn et al., 2024) enable model adaptation to a specific style via fine-tuning. Furthermore, certain approaches (Jeong et al., 2023; Hamazaspyan & Navasardyan, 2023; Wu et al., 2023; Hertz et al., 2023; Wang et al., 2024a; Yang et al., 2023; Chen et al., 2023) edit content and style in the U-Net's (Ronneberger et al., 2015) feature space, aiming to bypass style-tuning at the cost of reduced style transfer quality. Recently, Wang et al. (2023b); Liu et al. (2023); Sun et al. (2023); Qi et al. (2024) employ CLIP image encoder for extracting style features from each image. However, relying solely on semantic features extracted by a pre-trained CLIP image encoder as style features often results in poor style representation. Our study focuses on resolving these challenges by developing a specialized style-extracting encoder and producing the high-quality stylized images without test-time style-tuning.

## 3 METHOD

StyleShot is built on Stable Diffusion (Rombach et al., 2022), reviewed in Sec. 3.1. We first provide a brief overview of the pipeline for our method StyleShot, as illustrated in Fig. 3. Our pipeline comprises a style transfer model with a style-aware encoder (Sec. 3.2) and a content-fusion encoder (Sec. 3.3), as well as a style-balanced dataset StyleGallery along with a de-stylization (Sec. 3.4).

### 3.1 PRELIMINARY

Stable Diffusion consists of two processes: a diffusion process (forward process), which incrementally adds Gaussian noise $\epsilon$ to the data $x_0$ through a Markov chain. Additionally, a denoising process

generates samples from Gaussian noise $x_T \sim N(0, 1)$ with a learnable denoising model $\epsilon_\theta(x_t, t, c)$ parameterized by $\theta$. This denoising model $\epsilon_\theta(\cdot)$ is implemented with U-Net and trained with a mean-squared loss derived by a simplified variant of the variational bound:

$$\mathcal{L} = \mathbb{E}_{t, \mathbf{x}_0, \epsilon} \left[ \| \epsilon - \hat{\epsilon}_\theta(\mathbf{x}_t, t, c) \|^2 \right], \tag{1}$$

where $c$ denotes an optional condition. In Stable Diffusion, $c$ is generally represented by the text embeddings $f_t$ encoded from a text prompt using CLIP, and integrated into Stable Diffusion through a cross-attention module, where the latent embeddings $f$ are projected onto a query $Q$, and the text embeddings $f_t$ are mapped to both a key $K_t$ and a value $V_t$. The output of the block is defined as follows:

$$Attention(Q, K_t, V_t) = softmax \left( \frac{Q K_t^T}{\sqrt{d}} \right) \cdot V_t, \tag{2}$$

where $Q = W_Q \cdot f$, $K_t = W_{K_t} \cdot f_t$, $V_t = W_{V_t} \cdot f_t$ and $W_Q$, $W_{K_t}$, $W_{V_t}$ are the learnable weights for projection. In our model, the style embeddings are introduced as an additional condition and are amalgamated with the text's attention values.

## 3.2 STYLE-AWARE ENCODER

When training a style transfer model on a large-scale dataset where each image is considered a distinct style, previous methods (Liu et al., 2023; Wang et al., 2023b; Qi et al., 2024) often use CLIP image encoders to extract style features. However, CLIP is better at representing linguistic relevance to images rather than modeling image style, which comprises aspects like color, sketch, and layout that are difficult to convey through language, limiting the CLIP encoder's

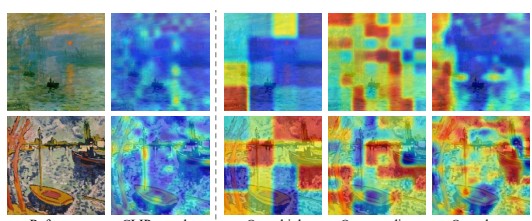

Figure 4: Attention map from the CLIP image encoder (left) and our style-aware encoder (right) on style reference images.

ability to capture relevant style features. As shown in Fig. 4 (left), the CLIP image encoder predominantly focuses on semantic information, often resulting in poor style representation. Therefore, we propose a style-aware encoder designed to specialize in extracting rich and expressive style embeddings.

**Style Extraction.** Our style-aware encoder borrows the pre-trained weights from CLIP image encoder, employing the transformer blocks to integrate the style information across patch embeddings. However, different from CLIP image encoder, which partitions the image into patches of a single scale following a single convolutional layer to learn the unified features, we adopt a multi-scale patch partitioning scheme in order to capture both low-level and high-level style cues. Specifically, we pre-process the reference image into non-adjacent patches $\mathbf{p_d}, \mathbf{p_m}, \mathbf{p_s}$ of three sizes—1/4, 1/8, and 1/16 of the image's length—with corresponding quantities of 8, 16, and 32, respectively. For these patches of three sizes, we use distinct ResBlocks of three depths $\mathcal{E}_d$, $\mathcal{E}_m$, and $\mathcal{E}_s$ as the MoE structure to separately extract patch embeddings $f_p$ at multiple level styles:

$$f_p = \left[ \mathcal{E}_d(\mathbf{p_d^1}); \cdots ; \mathcal{E}_d(\mathbf{p_d^8}); \mathcal{E}_m(\mathbf{p_m^1}); \cdots ; \mathcal{E}_m(\mathbf{p_m^{16}}); \mathcal{E}_s(\mathbf{p_s^1}); \cdots ; \mathcal{E}_s(\mathbf{p_s^{32}}) \right]$$

After obtaining multi-scale patch embeddings $f_p$ from varied-size patches, we employ a series of standard Transformer Blocks $\Phi$ for further style learning. To integrate the multiple level style features from $f_p$, we define a set of learnable style embeddings $f_s$, concatenated with $f_p$ as $[f_s, f_p]$, and feed $[f_s, f_p]$ into $\Phi$. This process yields expressive style embeddings $f_s$ with rich style representations from the output of $\Phi$:

$$[f_s, f_p] = \Phi \left( [f_s, f_p] \right)$$

Also, we drop the position embeddings to get rid of the spatial structure information in patches. Compared to methods based on the CLIP image encoder, which extracts semantic features from the single scale patch embeddings, our style-aware encoder provide more high-level style representations by featuring multi-scale patch embeddings. As shown in Fig. 4 (right), we visualized the attention maps for three distinct levels of patches in the style-aware encoder, our style-aware encoder does not solely focus on semantic areas but also style areas like the sky and water, which are often neglected by the CLIP image encoder.

**Style Injection.** Inspired by IP-Adapter (Ye et al., 2023), we infuse the style embeddings $f_s$ into a pre-trained Stable Diffusion model using a parallel cross-attention module. Specifically, similar to Eq. 2, we create an independent mapping function $W_{K_s}$ and $W_{V_s}$ to project the style embeddings $f_s$ onto key $K_s$ and value $V_s$. Additionally, we retain the query $Q$, projected from the latent embeddings $f$. Then the cross-attention output for the style embeddings is delineated as follows:

$$Attention(Q, K_s, V_s) = softmax\left(\frac{QK_s^T}{\sqrt{d}}\right) \cdot V_s, \tag{3}$$

the attention output of text embeddings $f_t$ and style embeddings $f_s$ are then combined as the new latent embeddings $f'$, which are then fed into subsequent blocks of Stable Diffusion:

$$f' = Attention(Q, K_t, V_t) + \lambda Attention(Q, K_s, V_s), \tag{4}$$

where $\lambda$ represents the weight balancing two components.

## 3.3 CONTENT-FUSION ENCODER

In practical scenarios, users provide text prompts or images as well as a style reference image to control the generated content and style, respectively. Previous methods (Jeong et al., 2023; Hertz et al., 2023) typically transfer style by manipulating content image features. However, the content features are coupled with style information, causing the generated images to retain the content's original style. This limitation hinders the performance of these methods in complex style transfer tasks. Differently, we pre-decouple the content information by eliminating the style information in raw image space, and then introduce a content-fusion encoder specifically designed for content and style integration.

**Content Extraction.** Currently, Wang et al. (2023a) utilizes de-colorization and subsequent DDIM Inversion (Song et al., 2020) for style removing. As demonstrated in Fig. 5 (a), this approach primarily targets low-level styles, leaving high-level styles like the brushwork of an oil painting and low poly largely intact. Edge detection algorithms such as Canny (Canny, 1986) and HED (Xie & Tu, 2015) can explicitly re-

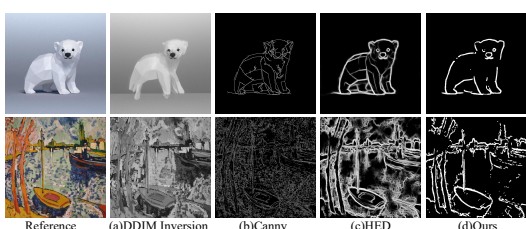

Reference  (a)DDIM Inversion  (b)Canny  (c)HED  (d)Ours

Figure 5: Illustration of the content input under different setting.

move style by generating a contour image. However, as illustrated in Figure 5 (b)(c), some high-level styles are still implicitly present in the edge details. To comprehensively remove the style from the reference image, we apply contouring using the HED Detector (Xie & Tu, 2015) along with thresholding and dilation. As a result, our content input $x_c$ (Fig. 5 (d)) remains only the essential content structure of the reference image.

Given the effectiveness of ControlNet in modeling spatial information within U-Net, we have adapted a similar structure for our content-fusion encoder. Specifically, our content-fusion encoder accepts content input $x_c$ as input, and outputs the latent representations for each layer as the content embeddings $f_c$:

$$f_c = \left[f_c^0, f_c^1, \cdot, f_c^L, \cdot\right],$$

where $f_c^0$ represents the latent representation of mid-sample block, $f_c^1, \cdot, f_c^L$ represent the latent representations of down-samples blocks and $L$ denotes the total number of layers in down-sample blocks. Moreover, we remove the text embeddings and employ style embeddings as conditions for the cross-attention layers within the content-fusion encoder to facilitate the integration of content and style.

**Content Injection.** Similar to ControlNet, we utilize a residual addition that strategically integrates content embeddings $f_c$ into the primary U-Net:

$$f^0 = f^0 + f_c^0,$$
$$f^i = f^i + f_c^{L-i+1}, i = 1, \cdot, L,$$

where $f^0$ represents the latent of mid-sample block in U-Net and $f^1$ to $f^L$ represent the latent representations of up-sample blocks in U-Net.

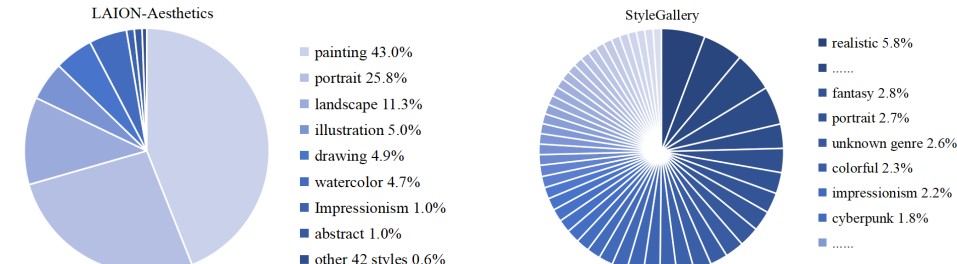

Figure 6: Style distribution analysis in LAION-Aesthetics (left) and our StyleGallery (right), the value represent the proportion of the top 50 styles in entire stylized data.

**Two-stage Training.** Given that the style embeddings are randomly initialized, jointly training the content and style components leads the model to reconstruct based on the spatial information from the content input, neglecting the integration of style embeddings in the early training steps. To resolve this issue, we introduce a two-stage training strategy. Specifically, we firstly train our style-aware encoder and corresponding cross-attention module while excluding the content component. This task fine-tuning on the whole style-aware encoder enables it to capture style relevant information. Subsequently, we exclusively train the content-fusion encoder with the frozen style-aware encoder.

### 3.4 StyleGallery & De-stylization

**StyleGallery.** Previous methods (Liu et al., 2023; Wang et al., 2023b) frequently utilized the LAION-Aesthetics (Schuhmann et al., 2022) dataset. Following the style analysis outlined in McCormack et al. (2024), we found that LAION-Aesthetics comprises only 7.7% stylized images. Further analysis revealed that the style images within LAION-Aesthetics are characterized by a pronounced long-tail distribution. As illustrated in Fig. 6, painting style accounts for 43% of the total style samples while the combined proportion of other 42 styles is less than 0.6%. Models trained on the dataset with extremely imbalanced distribution easily overfit to high-frequency styles, which compromises their ability to generalize to rare or unseen styles, as detailed in the experimental analysis in Sec. 4.4. This indicates that the efficacy of style transfer is closely associated with the style distribution of the training dataset.

Motivated by this observation, we construct a style-balanced dataset, called StyleGallery, covering several open source datasets. Specifically, StyleGallery includes JourneyDB Sun et al. (2024), a dataset comprising a broad spectrum of diverse styles derived from MidJourney, and WIKIART Phillips & Mackintosh (2011), with extensive fine-grained painting styles, such as pointillism and ink drawing, and a subset of stylized images from LAION-Aesthetics. 99.7% of the images in our StyleGallery have style descriptions. The style distribution within StyleGallery is more balanced and diverse as illustrated in Fig. 6, which benefits our model in learning expressive and generalized style representation.

**De-stylization.** We notice that the text prompts for images frequently contain detailed style descriptions, such as "a movie poster for The Witch *in the style of Arthur rackham*", leading to the entanglement of style information within both text prompt and reference image. Since the pre-trained Stable Diffusion model is well responsive to text conditions, such an entanglement may hinder the model's ability to learn style features from the reference image. Consequently, we endeavor to remove all style-related descriptions from the text across all text-image pairs in StyleGallery, retaining only content-related text. Our decoupling training strategy separates style and content information into distinct inputs, aiming to improve the extraction of style embeddings from StyleGallery.

## 4 Experiments

### 4.1 Style Evaluation Benchmark

Previous works (Liu et al., 2023; Ruiz et al., 2023; Sohn et al., 2024; Wang et al., 2023b) established their own evaluation benchmarks with limited style images which are not publicly available. To comprehensively evaluate the effectiveness and generalization ability of style transfer methods, we build StyleBench that covers 73 distinct styles, ranging from paintings, flat illustrations, 3D rendering to sculptures with varying materials. For each style, we collect 5-7 distinct images with variations.

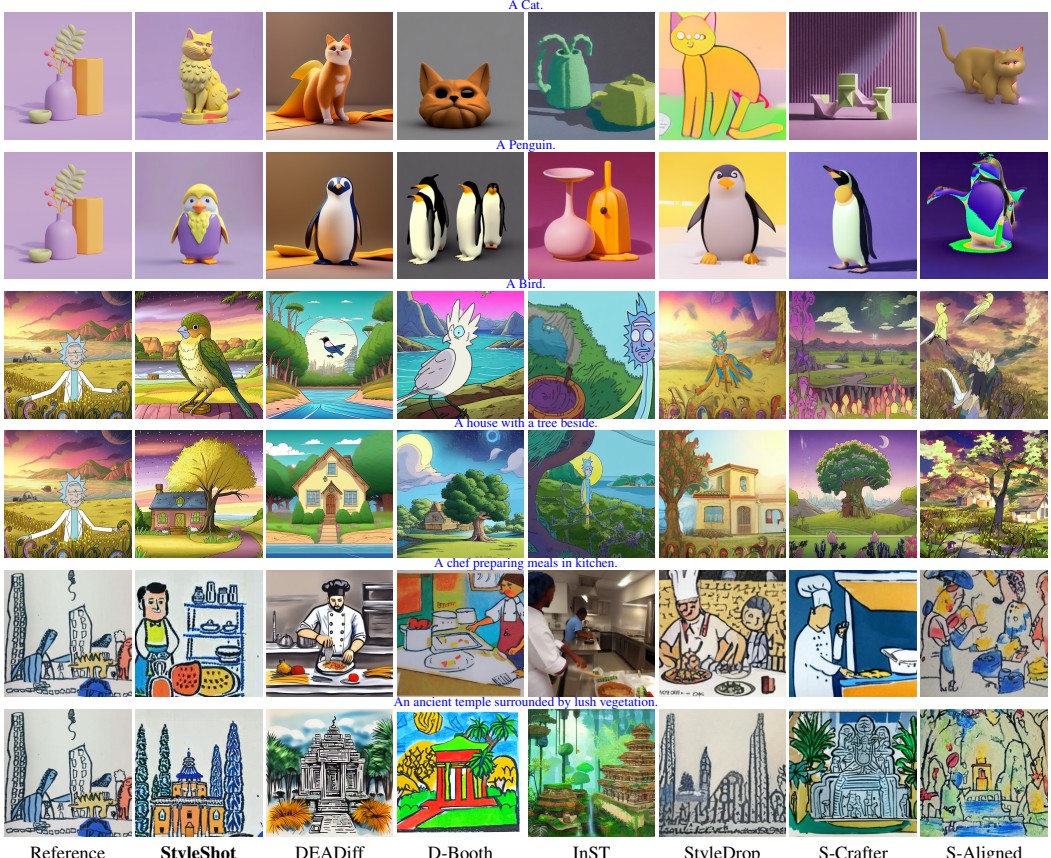

A Cat.

A Penguin.

A Bird.

A house with a tree beside.

A chef preparing meals in kitchen.

An ancient temple surrounded by lush vegetation.

| Reference | **StyleShot** | DEADiff | D-Booth | InST | StyleDrop | S-Crafter | S-Aligned |

Figure 7: Qualitative comparison with SOTA text-driven style transfer methods.

In total, our StyleBench contains 490 images across diverse styles. Moreover, we generated 20 text prompts and 40 content images from simple to complex that describe random objects and scenarios as content input. Details are available in the Appendix A. We conduct qualitative and quantitative comparisons on this benchmark.

## 4.2 QUALITATIVE RESULTS

**Text-driven Style Learning.** Fig. 1 has displayed results of StyleShot to six distinct style images, each corresponding to the same pair of textual prompts. For fair comparison, we also present results of other text-driven style transfer methods, such as DEADiff (Qi et al., 2024), DreamBooth (Ruiz et al., 2023) on Stable Diffusion, InST (Zhang et al., 2023), StyleDrop (Sohn et al., 2024) (unofficial implementation), StyleCrafter (Liu et al., 2023) and StyleAligned (Hertz et al., 2023) applied to three style reference images, with two different text prompts for each reference image. As shown in Fig. 7, we observe that StyleShot effectively captures a broad spectrum of style features, ranging from basic elements like colors and textures to intricate components like layout, structure, and shading, resulting in a desirable stylized imaged aligned to text prompts. This shows the effectiveness of our style-aware encoder to extract rich and expressive style embeddings.

Furthermore, we train StyleCrafter, a style transfer method adopting a frozen CLIP-based encoder, on StyleGallery to extract style representations. As illustrated in Fig. 10, setting default scale value $\lambda = 1$ during inference on StyleCrafter results in significant content leakage issue while setting the scale value $\lambda = 0.5$ diminished the style injection, generating even some real-world images. Conversely, our StyleShot generates the stylized images align with the text prompt and style reference. Beyond its effective style and text alignment, StyleShot also demonstrates the capacity to discern and learn fine-grained stylistic details as shown in Fig. 9. More different baseline comparisons and visualizations are available in Appendix B.3 and B.4.

**Image-driven Style Learning.** Thanks to our content-fusion encoder, StyleShot also excels at transferring style onto content images. We compare StyleShot with other SOTA image-driven style transfer methods such as AdaAttN (Liu et al., 2021), EFDM (Zhang et al., 2022a), StyTR-2 (Deng

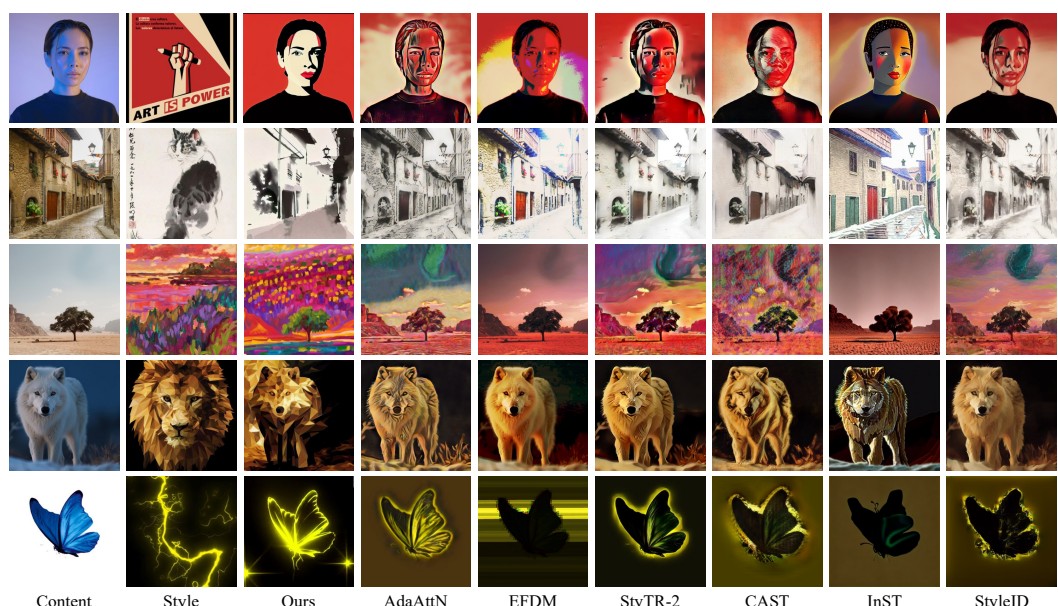

Figure 8: Qualitative comparison with SOTA image-driven style transfer methods.

et al., 2022), CAST (Zhang et al., 2022b), InST (Zhang et al., 2023) and StyleID (Chung et al., 2024). As illustrated in Fig. 8, our StyleShot can transfer any style (including even complex and high-level styles such as light, pointillism, low poly, and flat) onto various content images (such as humans, animals, and scenes), while baseline methods excel primarily in painting styles and struggle with these high-level styles. This shows the efficacy of the content-fusion encoder in achieving superior style transfer performance while maintaining the structural integrity of the content image.

Table 1: Quantitative comparison from human preference style loss and clip scores on text and image alignment with SOTA text-driven style transfer methods. Best result is marked in **bold**.

| Metrics | StyleCrafter | DEADiff | StyleDrop | InST | StyleAligned | StyleShot |
|---|---|---|---|---|---|---|
| human text ↑ | 0.097 | 0.193 | 0.060 | 0.127 | 0.080 | **0.443** |
| human image ↑ | 0.143 | 0.080 | 0.040 | 0.063 | 0.173 | **0.500** |
| clip text ↑ | 0.202 | **0.232** | 0.220 | 0.204 | 0.213 | 0.219 |
| clip image ↑ | **0.706** | 0.597 | 0.621 | 0.623 | 0.680 | 0.640 |
| style loss ↓ | 9.704 | 30.869 | 12.327 | 14.440 | 15.454 | **8.691** |

## 4.3 QUANTITATIVE RESULTS

**Human Preference.** Following Liu et al. (2023); Wang et al. (2023b); Sohn et al. (2024), we conduct user preference study to evaluate the text and style alignment ability on text-driven style transfer. Results are tabulated in Tab. 1. Compared to other methods, our StyleShot achieves the highest text/style alignment scores with a large margin, demonstrating the robust stylization across various styles and responsiveness to text prompts.

Table 2: Quantitative comparison from clip image score and style loss with SOTA image-driven style transfer methods. Best result is marked in **bold**.

| Metrics | AdaAttN | EFDM | StrTR-2 | CAST | InST | StyleID | StyleShot |
|---|---|---|---|---|---|---|---|
| clip image ↑ | 0.569 | 0.561 | 0.586 | 0.575 | 0.569 | 0.604 | **0.660** |
| style loss ↓ | 6.654 | 22.003 | **5.228** | 9.439 | 6.645 | 10.295 | 7.872 |

**Other Metrics.** Following Wang et al. (2023a), we also measure the clip scores (Radford et al., 2021) and style loss (Gatys et al., 2016; Huang & Belongie, 2017). As shown in Tab. 1 and Tab. 2, StyleShot achieves the best clip image score and style loss in image-driven and text-driven style transfer settings, respectively. However, as previously mentioned in Sohn et al. (2024); Liu et al. (2023) and Wang et al. (2023a), CLIP scores and style loss are not ideal for evaluation in style transfer tasks. We present these evaluation results for reference purposes only.

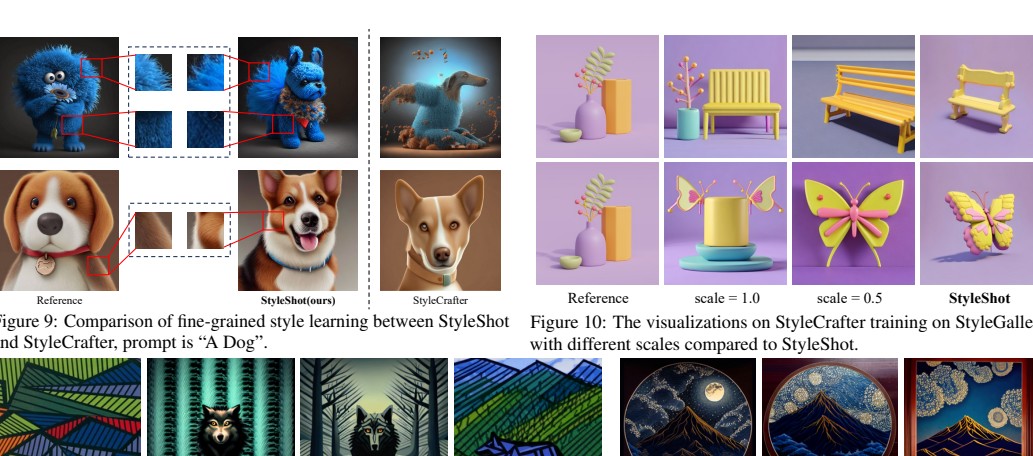

Figure 9: Comparison of fine-grained style learning between StyleShot and StyleCrafter, prompt is "A Dog".

Figure 10: The visualizations on StyleCrafter training on StyleGallery with different scales compared to StyleShot.

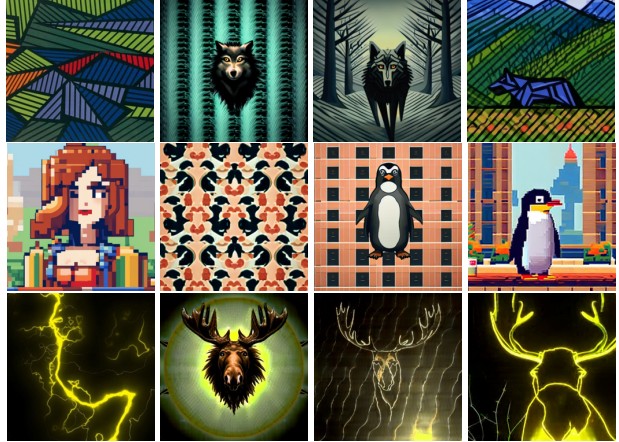

Figure 11: The visualizations on multi-level style extraction, from top to bottom prompts are "A wolf walking stealthily through the forest", "A penguin", "A moose".

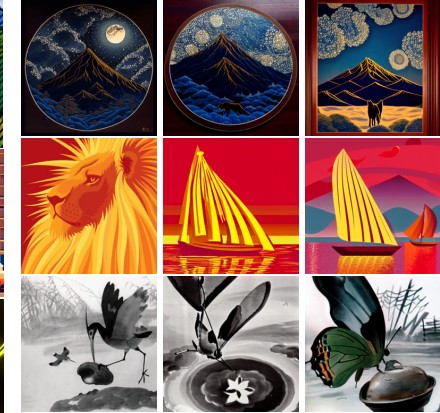

Figure 12: The visual illustration of StyleCrafter training on our StyleGallery and Laion-Aesthetics dataset, from top to bottom prompts are "A wolf walking stealthily through the forest", "A wooden sailboat docked in a harbor", "A colorful butterfly resting on a flower".

## 4.4 ABLATION STUDIES

**Style-aware Encoder.** By selectively dropping patch embeddings of varying sizes, we verified the style-aware encoder's ability to extract style features at multiple levels. As illustrated in Fig. 11, retaining only the smallest patches results in generating images that solely inherit low-level style information, such as color. However, when larger-sized patches are included, the generated images begin to exhibit more high-level style.

Moreover, we utilize a frozen CLIP image encoder without multi-scale patch embeddings as a basline. We then apply task fine-tuning and multi-scale patch embeddings to this baseline model. As shown in Fig. 13, the style extracted by the baseline is notably different from the reference. After including task fine-tuning and multi-scale patch embeddings, the style of reference image is better captured by the model. These results demonstrate the effectiveness of incorporating both task fine-tuning and multi-scale patch embeddings in the style encoder to extract more expressive and richer style representations. More experiments are shown in Appendix B.5.

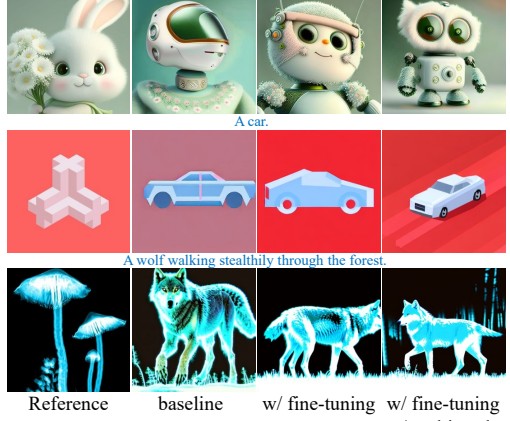

Figure 13: Visualizations incorporating task fine-tuning and a multi-scale patch embeddings in the CLIP image encoder.

**Content-fusion Encoder.** To evaluate the content-fusion encoder, we integrated pre-trained Control-Net models (conditioned on Canny, HED, and our content input) with our style-aware encoder on Stable Diffusion. As illustrated in Fig. 14, compared to Canny and HED, our content input enabled greater stylization, demonstrating the efficacy of our contouring technique for content decoupling. Moreover, we train the content-fusion encoder with our style-aware encoder. By incorporating style

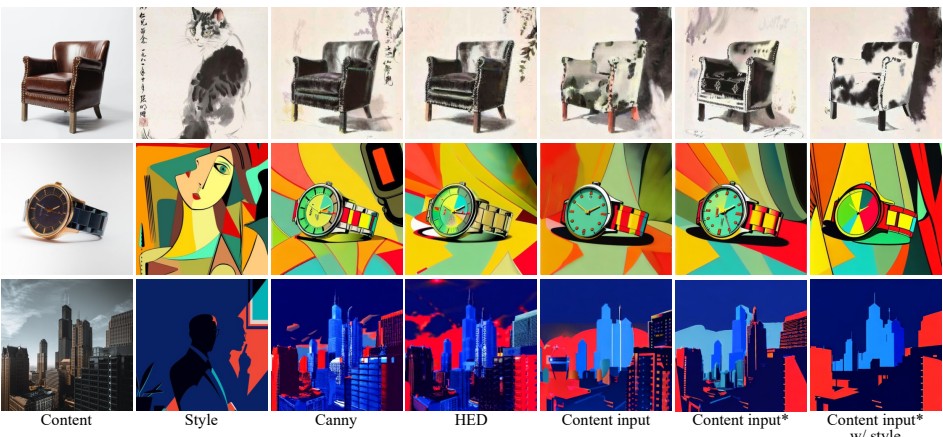

Figure 14: Ablation studies on our content-fusion encoder. Columns 3-5 integrate the pre-trained ControlNet. * represents training content-fusion encoder with style-aware encoder.

embeddings into the content-fusion encoder, the combination of style and content becomes more smooth, demonstrating the effectiveness of our content-fusion encoder.

**Style-balanced Dataset.** We conduct ablations by respectively training models on the LAION-Aesthetics and JourneyDB datasets. As shown in Tab. 3, model trained on StyleGallery achieves the highest image alignment scores. Visual analysis in Fig. 15 indicates that the model trained on StyleGallery effectively recognizes and generate a butterfly in the pointillism style. Moreover, as depicted in Fig. 12, images generated by StyleCrafter trained on our StyleGallery also exhibit superior style alignment with the reference image. This underscores the importance of utilizing a style-balanced dataset for training style transfer methods. More experiments are shown in Appendix B.6.

Table 3: Image alignment scores on various datasets.

| Dataset | LAION | JourneyDB | StyleGallery |
|---------|-------|-----------|--------------|
| image ↑ | 0.614 | 0.618 | **0.640** |

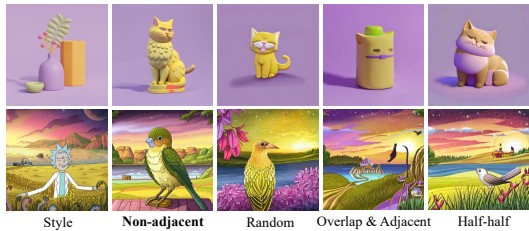

Reference    LAION-Aes.    JourneyDB    StyleGallery
Figure 15: Visualization of Tab. 3, "a butterfly".

**Partitioning Strategy.** We employ a non-adjacent partitioning strategy (See Appendix Fig. 19) to disrupt the structural information of the image so as to reduce the content leakage issue. To validate the effectiveness of this partitioning strategy, we implement three other typical partitioning strategies as follow: 1) Random: Patches are randomly cropped from the image; 2) Overlap & adjacent: Patches are uniform sampled in overlapped and adjacent way. 3) Half-half: Largest patches are cropped from the left half of the image and the remaining patches from the right half. As shown in Fig. 16, our partitioning (row 2) yields more stable and superior style transfer performance compared to the Random (row 3) and Half-half (row 5). Furthermore, the Overlap & Adjacent partitioning (row 4) leads to content leakage and fails to respond adequately to textual input.

Style    **Non-adjacent**    Random    Overlap & Adjacent    Half-half
Figure 16: Visual results of different partitioning strategies.

## 5 CONCLUSION

In this paper, we introduce StyleShot, the first work to specially designate a style-aware encoder to extract rich style in style transfer task on diffusion model. StyleShot can accurately identify and transfer the style of any reference image without test-time style-tuning. Particularly, due to the design of the style-aware encoder, which is adept at capturing style representations, StyleShot is capable of learning an expressive style such as shading, layout, and lighting, and can even comprehend fine-grained style nuances. With our content-fusion encoder, StyleShot achieves remarkable performance in image-driven style transfer. Furthermore, we identified the beneficial effects of stylized data and developed a style-balanced dataset StyleGallery to improve style transfer performance. Extensive experimental results validate the effectiveness and superiority of StyleShot over existing methods.

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

Figure R1: Visual results for different content extractions on ControlNet or our proposed content-fusion encoder (*).

Figure R2: Visual comparison among StyleShot, DreamStyler, Z-Star and Master.

Figure R3: Visual result with or without de-stylization.

Figure R4: Qualitative comparison with SOTA text-driven style transfer methods on third-party bench from [1].

[1]. "A Comprehensive Evaluation of Arbitrary Image Style Transfer Methods'', IEEE TVCG.

Figure R5: Qualitative comparison with SOTA image-driven style transfer methods on third-party bench from [1].

# APPENDIX / SUPPLEMENTAL MATERIAL

## A    STYLE EVALUATION BENCHMARK

### A.1    STYLE IMAGES

In this section, we provide more details about our style evaluation benchmark, called StyleBench. We collect images in StyleBench from the Internet. The 73 types of styles in StyleBench are as shown in the Tab. 4.

Table 4: 73 style types in StyleBench.

| 3D Model 00/.../05 | Abstract 00/01 | Analog film | Anime 00/.../07 | Art deco |
|---|---|---|---|---|
| Baroque | Children's Painting | Classicsm | Constructivism | Craft Clay |
| Cublism | Cyberpunk | Expressionist | Fantasy Art | Fauvism |
| Flat Vector | Folk art | Gongbi | Graffiti | Hyperrealism |
| Icon 00/01/02 | Impressionism | Ink and Wash Painting | IsoMetric | Japonism |
| Line Art | Low Poly | Luminism | Macabre | MineCraft |
| Monochrome | Neoclassicism | Neo-Figurative Art | Nouveau | Op Art |
| Origami | Orphism | Photographic | Pixel Art | Pointilism |
| Pop Art | Post-Impressionism | Precisionism | Primitivism | Psychedelic |
| Realism | Rococo | Smoke & Light | Statue | Steampunk |
| Stickers | Stick Figure | Surrealist | Symbolism | Tonalism |
| Typography | Watercolor | others | | |

Among these, due to the variations in fine-grained style features, categories such 3D models, Anime, Icons, and Stick Figures can be subdivided into more specific groups. For these subdivisions, we employ numerical labels for further classification, for example, 3D Model 00 through 05. As depicted in Fig. 17, each style comprises six to seven images, amounting to a total of 490 style images in our evaluation benchmark.

Table 5: 20 text prompts in StyleBench.

| "A bench" | "A bird" | "A butterfly" | "An elephant" |
|---|---|---|---|
| "A car" | "A dog" | "A cat" | "A laptop" |
| "A moose" | "A penguin" | "A robot" | "A rocket" |
| "An ancient temple surrounded by lush vegetation" | | | |
| "A chef preparing meals in kitchen" | | | |
| "A colorful butterfly resting on a flower" | | | |
| "A house with a tree beside" | | | |
| "A person jogging along a scenic trail" | | | |
| "A student walking to school with backpack" | | | |
| "A wolf walking stealthily through the forest" | | | |
| "A wooden sailboat docked in a harbor" | | | |

### A.2    TEXT PROMPTS

We have collected 20 text prompts, as shown in Tab. 5. Our text prompts employ sentences that vary from simple to complex in order to depict a diverse array of objects and character images.

### A.3    CONTENT IMAGES

We have collected 40 content images, as shown in Fig. 18.

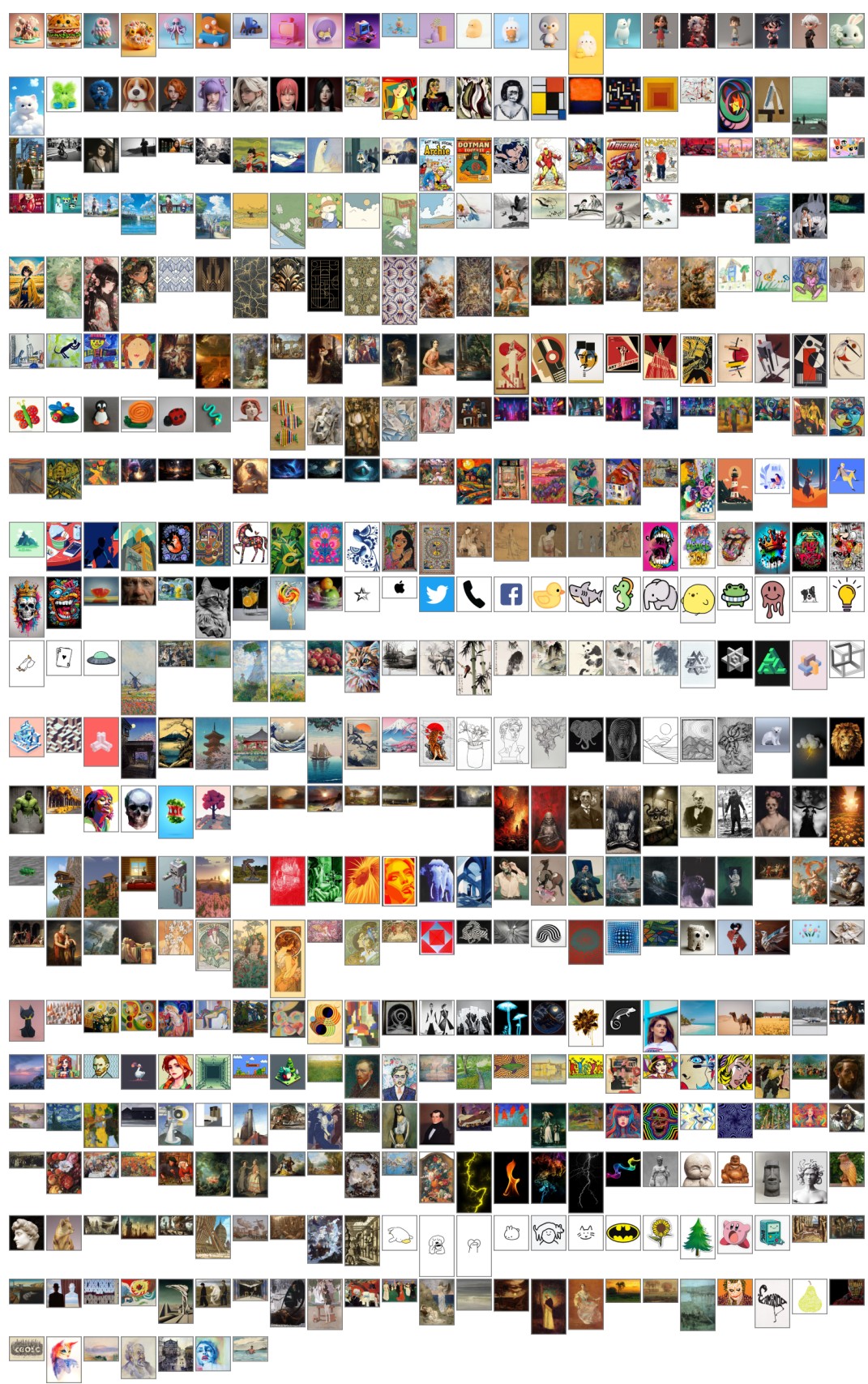

Figure 17: 490 style images in StyleBench.

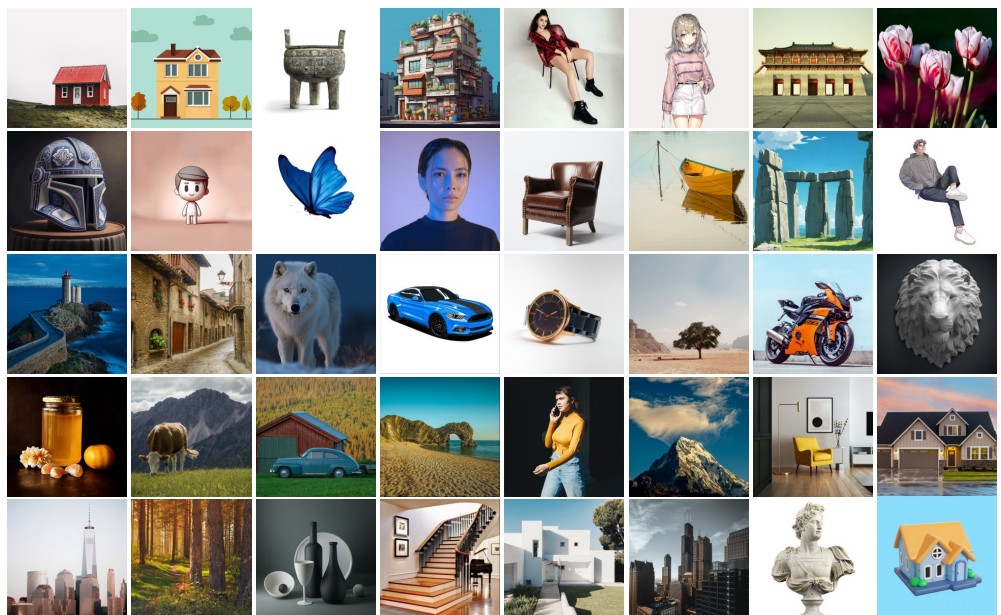

Figure 18: 40 content images in StyleBench.

## B EXPERIMENTS

### B.1 IMPLEMENTATION DETAILS

In this section, we first provide some implementation details about our style-aware encoder discussed in Sec 3.2. We adopt the open-sourced SD v1.5 as our base T2I model. We construct our StyleGallery with diverse styles, which totally contain 5.7M image-text pairs, including open source datasets such as JourneyDB, WiKiArt and a subset of stylized images from LAION-Aesthetics. Our varied-size patches are divided into three sizes 1/4, 1/8 and 1/16 of image length with corresponding quantities of 8, 16, and 32, as shown in Fig. 19. For patches of varying sizes, we utilize ResBlocks with differing depths implemented using six, five, and four ResBlocks, respectively. Furthermore, our Transformer Blocks are initialized from the pre-trained weights of OpenCLIP ViT-H/14 (Ilharco et al., 2021).

Following the Transformer Blocks, we introduce an additional MLP for the style embeddings. Similar to IP-Adapter, in each layer of the diffusion model, a parallel cross-attention module is utilized to incorporate the projected style embeddings. We train our StyleShot on a single machine with 8 A100 GPUs for 360k steps (300k for stage one, 60k for stage two) with a batch size of 16 per GPU, and set the AdamW optimizer (Loshchilov & Hutter, 2017) with a fixed learning rate of 0.0001 and weight decay of 0.01. During the training phase, the shortest side of each image is resized to 512, followed by a center crop to achieve a $512 \times 512$ resolution. Then the image is sent to the U-Net as the target image and to the Style-Aware encoder as the reference image. To enable classifier-free guidance, text and images are dropped simultaneously

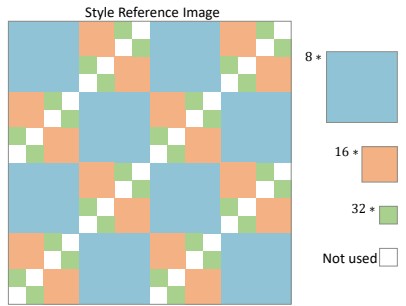

Figure 19: Illustration of partitioning our style reference image.

with a probability of 0.05, and images are dropped individually with a probability of 0.25. During the inference phase, we adopt PNDM (Liu et al., 2022) sampler with 50 steps, and set the guidance scale to 7.5 and $\lambda = 1.0$.

### B.2 DETAILS ON HUMAN PREFERENCE

In this section, we provide details about the human preference study discussed in Sec. 4.3. We devised 30 tasks to facilitate comparisons among StyleDrop (Sohn et al., 2024), StyleShot (ours), StyleAligned (Hertz et al., 2023), InST (Zhang et al., 2023), StyleCrafter (Liu et al., 2023) and

DEADiff (Qi et al., 2024) with each task including a reference style image, text prompt, and a set of six images for assessment by the evaluators. We describe detailed instruction for each task, and ultimately garnered 1320 responses.

**Instruction.**

In our study, we evaluated 30 tasks, each involving a reference style image and the images generated by six distinct text-driven style transfer algorithms. Participants are required to select the generated image that best matches based on two criteria:

- Style Consistency: The style of the generated image aligns with that of the reference style image;
- Text Consistency: The depicted content of generated image correspond with the textual description;

**Questions.**

- Which generated image best matches the style of the reference image? Image A, Image B, Image C, Image D, Image E, Image F.
- Which generated image is best described by the text prompt? Image A, Image B, Image C, Image D, Image E, Image F.

## B.3 EXTENDED BASELINE COMPARISONS

In this section, we first provide additional qualitative comparison with other SOTA text-driven style transfer methods Dreamstyler (Ahn et al., 2024), T2I-Adapter (Mou et al., 2024), IP-Adapter (Ye et al., 2023) and InstantStyle(including sdxl version) (Wang et al., 2024a) in Fig. 20. We also provide the quantitative results of these baselines, as shown in Tab. 6. StyleShot achieves superior performance compared to these methods. We also observe that these methods have serious content leakage issue, which leads to a high clip image score and a very low clip text score.

Table 6: Quantitative comparison from style loss and clip scores on text and image alignment with *other* SOTA text-driven style transfer methods. Best result is marked in **bold**.

| Metrics | Dreamstyle | T2I-Adapter | IP-Adapter | InstantStyle | InstantStyle(sdxl) | StyleShot |
|---|---|---|---|---|---|---|
| clip text ↑ | 0.189 | 0.133 | 0.207 | 0.127 | 0.212 | **0.219** |
| clip image ↑ | 0.638 | **0.771** | 0.714 | 0.761 | 0.684 | 0.640 |
| style loss ↓ | 19.273 | 13.512 | 10.147 | 8.721 | 8.744 | **8.691** |

Moreover, we provide more qualitative comparison with SOTA text-driven style transfer methods StyleDrop (Sohn et al., 2024), DEADiff (Qi et al., 2024), InST (Zhang et al., 2023), Dream-Booth (Ruiz et al., 2023), StyleCrafter (Liu et al., 2023), StyleAligned (Hertz et al., 2023) in Fig. 25 and SOTA image-driven style transfer methods AdaAttN (Liu et al., 2021), EFDM (Zhang et al., 2022a), StyTR-2 (Deng et al., 2022), CAST (Zhang et al., 2022b), InST (Zhang et al., 2023) and StyleID (Chung et al., 2024) in Fig. 26.

## B.4 EXTENDED VISUALIZATION

In this section, we present additional text-driven style transfer visualization results for StyleShot across various styles, as shown in Fig. 27, 28. Unlike Fig. 25, each row in Fig. 27, 28 displays stylized images within a specific style, where the first column represents the reference style image, and the next six columns represent images generated under that style with distinct prompts. We also present the additional experiments image-driven style transfer visualization results for StyleShot across various styles, as shown in Fig. 29.

## B.5 STYLE-AWARE ENCODER.

In this section, we provide quantitative evaluations for the experiments discussed in Sec. 4.4 paragraph *style-aware encoder* in Tab. 7 and Tab. 8.

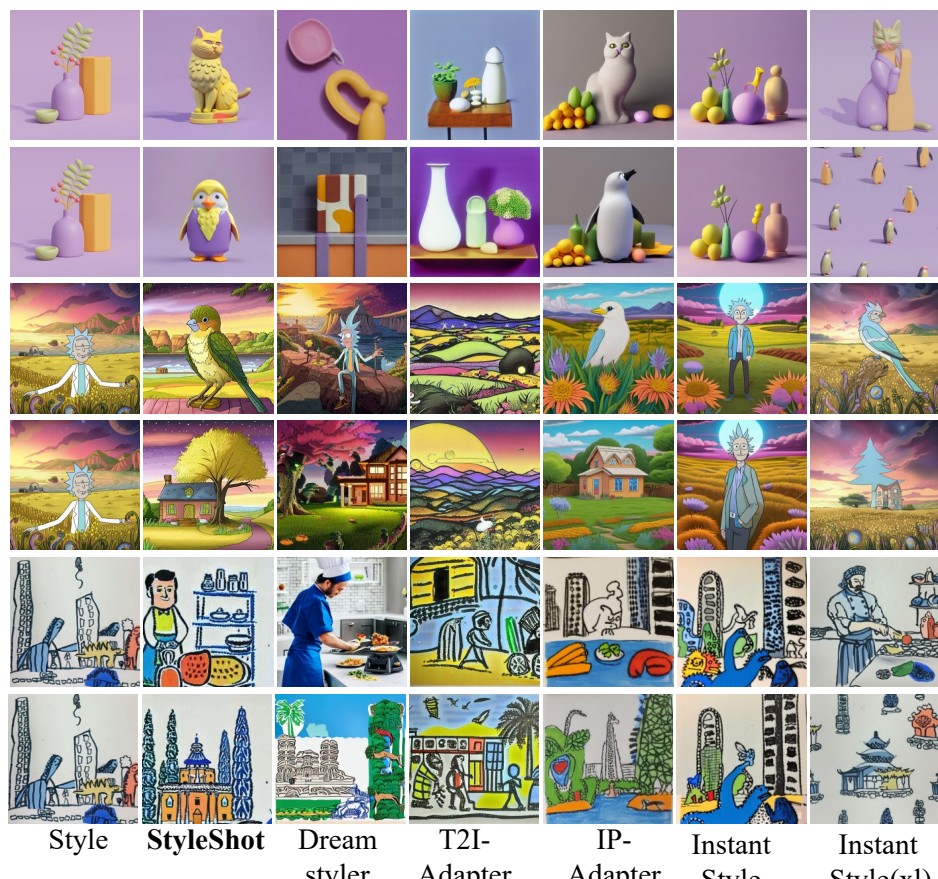

| | | | | | | |
|---|---|---|---|---|---|---|
| Style | **StyleShot** | Dream styler | T2I-Adapter | IP-Adapter | Instant Style | Instant Style(xl) |

Figure 20: Qualitative comparison with *other* SOTA image-driven style transfer methods.

Table 7: Quantitative comparison between StyleCrafter and StyleShot. Best result is marked in **bold**.

| Metrics | StyleCrafter(scale=1.0) | StyleShot |
|---|---|---|
| clip image ↑ | **0.780** | 0.640 |
| clip text ↑ | 0.149 | **0.219** |
| style loss ↓ | 18.443 | **8.691** |

As shown in Fig. 10, we observe that StyleCrafter (scale=1.0) has content leakage issue, with a high clip image score and a low clip text score. According to the visual results in Fig. 11, maintaining patches of all sizes in the style-aware encoder yields the best quantitative results.

Table 8: Quantitative comparison between multi-level partitioning. Best result is marked in **bold**.

| Metrics | Low | Low, Medium | Low, Medium, High |
|---|---|---|---|
| clip image ↑ | 0.549 | 0.586 | **0.640** |
| clip text ↑ | 0.168 | 0.208 | **0.219** |
| style loss ↓ | 130.652 | 19.062 | **8.691** |

Moreover, we visualized the distribution of attention weights for three levels of patches in Fig. 21. Medium level patches receive the highest attention weights.

## B.6 STYLE-BALANCED DATASET.

A large-scale style-balanced dataset is valuable for learning representative style features, enabling effective generalization to unseen styles. The underlying reason is that models tend to learn low-level style features such as color, texture if the majority of the training dataset is real-world images. Consequently, it is difficult for the models to recognize high-level style features such as styles

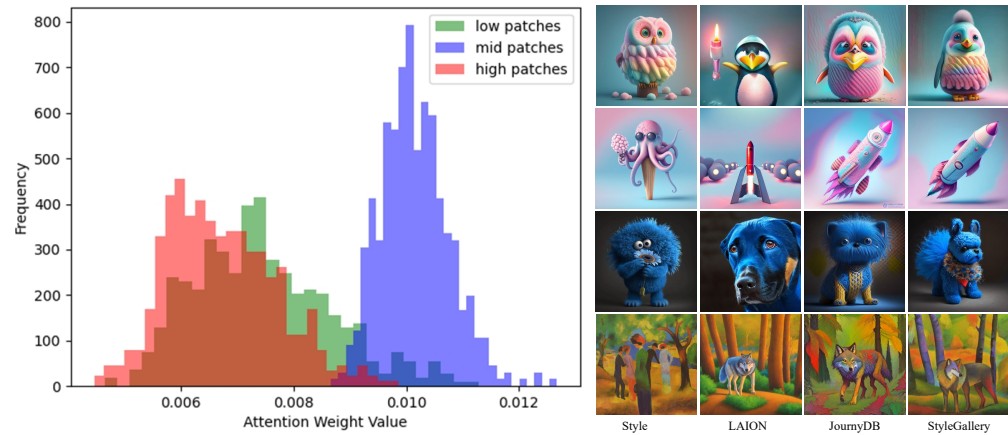

Figure 21: Attention weights distributions of StyleShot. Figure 22: More style images from StyleShot trained on different datasets.

expressed via layout, light, line art, the ones not embodied by real-world images. In this section, we provide more visual examples on training StyleShot on different dataset in Fig. 22.

### B.7 MULTI-SHOT STYLE TRANSFER.

StyleShot is compatible with the multi-shot style transfer by averaging style embeddings. As demonstrated in Fig. 23, StyleShot shows stable multi-shot style transfer capability.

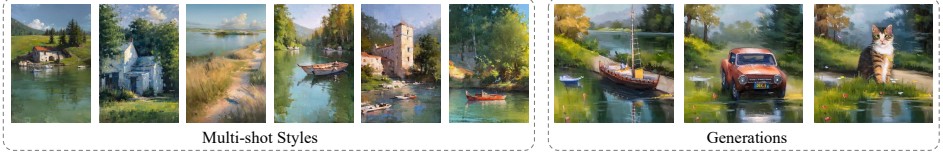

Figure 23: Multi-shot style transfer result of StyleShot.

### B.8 DIFFERENT STYLE GUIDANCE SCALE.

As shown in Fig. 24, we show the results that try different style guidance scale $\lambda$ for the same input at test time of StyleShot.

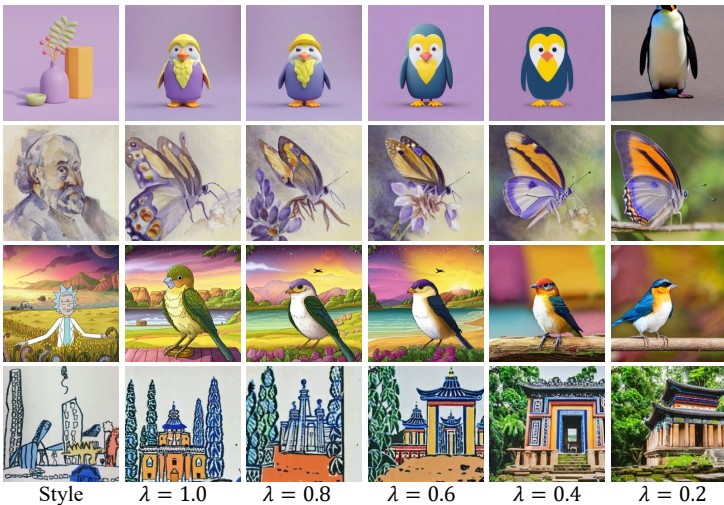

Figure 24: Visual results for different style guidance scale for the same input at test time.

## B.9 DE-STYLIZATION.

In Sec. 3.4, we removed the style descriptions in the text prompt to decouple style and content into the reference images and text prompts during training. To validate the effectiveness of this de-stylization, we trained the model with text prompts that did not have the style descriptions removed. The quantitative results in Tab. 9 indicate that the style descriptions in the text can adversely impact our model's learning of the style to some extent.

Table 9: De-stylization on prompts.

| Prompt | With Style | De-Style |
|---|---|---|
| image ↑ | 0.631 | **0.640** |

## B.10 RUNNING TIME COST ANALYSIS.

In this section, we provide the running time cost analysis with StyleShot and other SOTA style transfer methods StyleDrop (Sohn et al., 2024), DEADiff (Qi et al., 2024), InST (Zhang et al., 2023), Dream-Booth (Ruiz et al., 2023), StyleCrafter (Liu et al., 2023), StyleAligned (Hertz et al., 2023), as shown in Tab. 10. Firstly, for StyleShot, DEADiff and StyleCrafter, once training is complete, the test running time depends solely on the diffusion inference process. Conversely, style-tuning methods such as Dream-Booth (500 steps), StyleDrop (1000 steps) and InsT(6100 steps) require additional time for tuning reference style images. Furthermore, StyleAligned shares the self-attention of the reference image during inference, necessitating an inversion process. It should be noted that all diffusion-based methods have their inference steps set to 50, and we have calculated the running time cost for a single image on a A100 GPU.

Table 10: Running time cost between StyleShot and others SOTA style transfer methods.

| TYPE | DEADiff | D-Booth | S-Crafter | StyleDrop | InST | S-Aligned | **StyleShot** |
|---|---|---|---|---|---|---|---|
| training | - | 371s | - | 302s | 1868s | - | - |
| inference | 3s | 5s | 5s | 7s | 5s | 18s | 5s |

## C LIMITATIONS & DISCUSSIONS.

In this paper, we highlight that a style-aware encoder, specifically designed to extract style embeddings, is beneficial for style transfer tasks. However, we have not explored all potential designs of the style encoder, which warrants further investigation.

## D LICENSE OF ASSETS

The adopted JourneyDB dataset (Sun et al., 2024) is distributed under `https://journeydb.github.io/assets/Terms_of_Usage.html` license, and LAION-Aesthetics (Schuhmann et al., 2022) is distributed under MIT license. We implement the model based on IP-Adapter codebase (Ye et al., 2023) which is released under the Apache 2.0 license.

We will publicly share our code and models upon acceptance, under Apache 2.0 License.

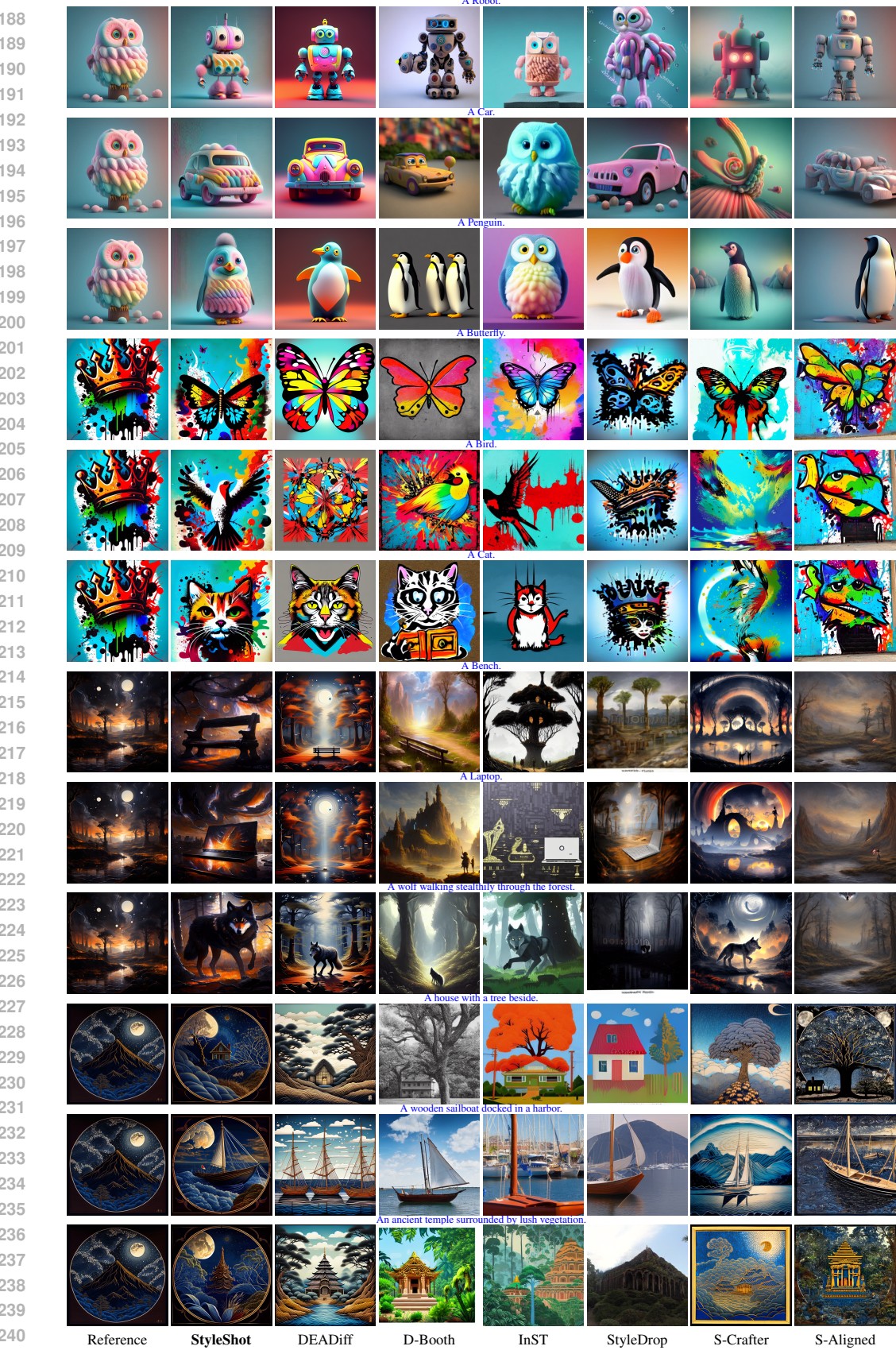

Figure 25: Qualitative comparisons with SOTA text-driven style transfer methods.

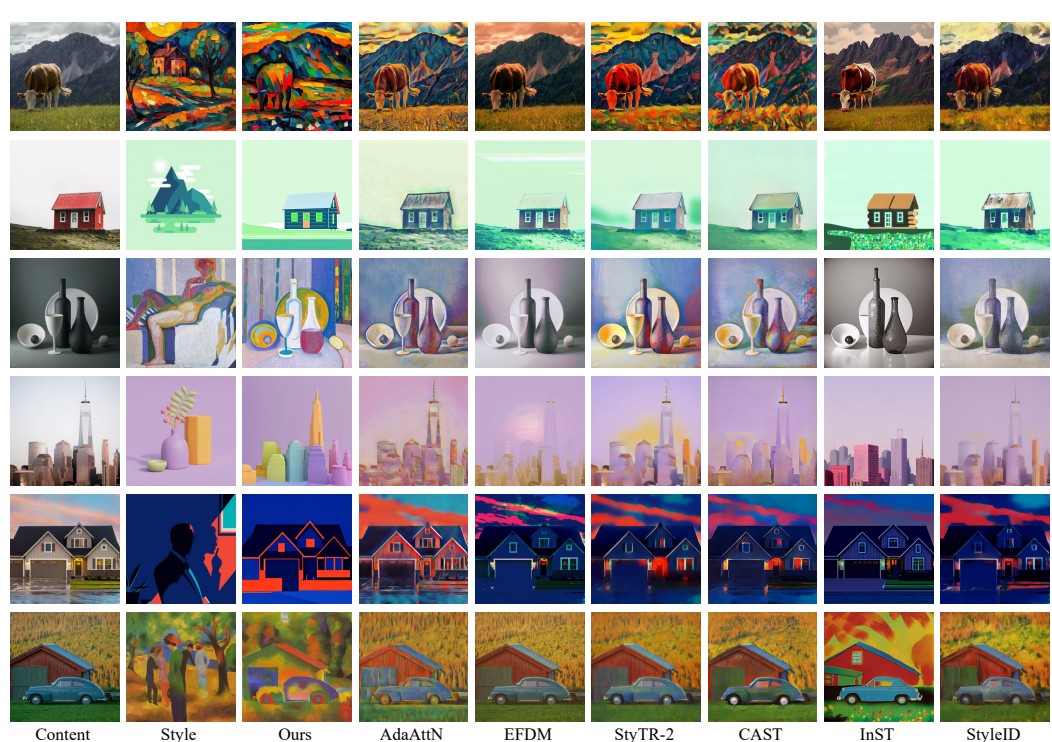

Content    Style    Ours    AdaAttN    EFDM    StyTR-2    CAST    InST    StyleID

Figure 26: Qualitative comparisons with SOTA image-driven style transfer methods.

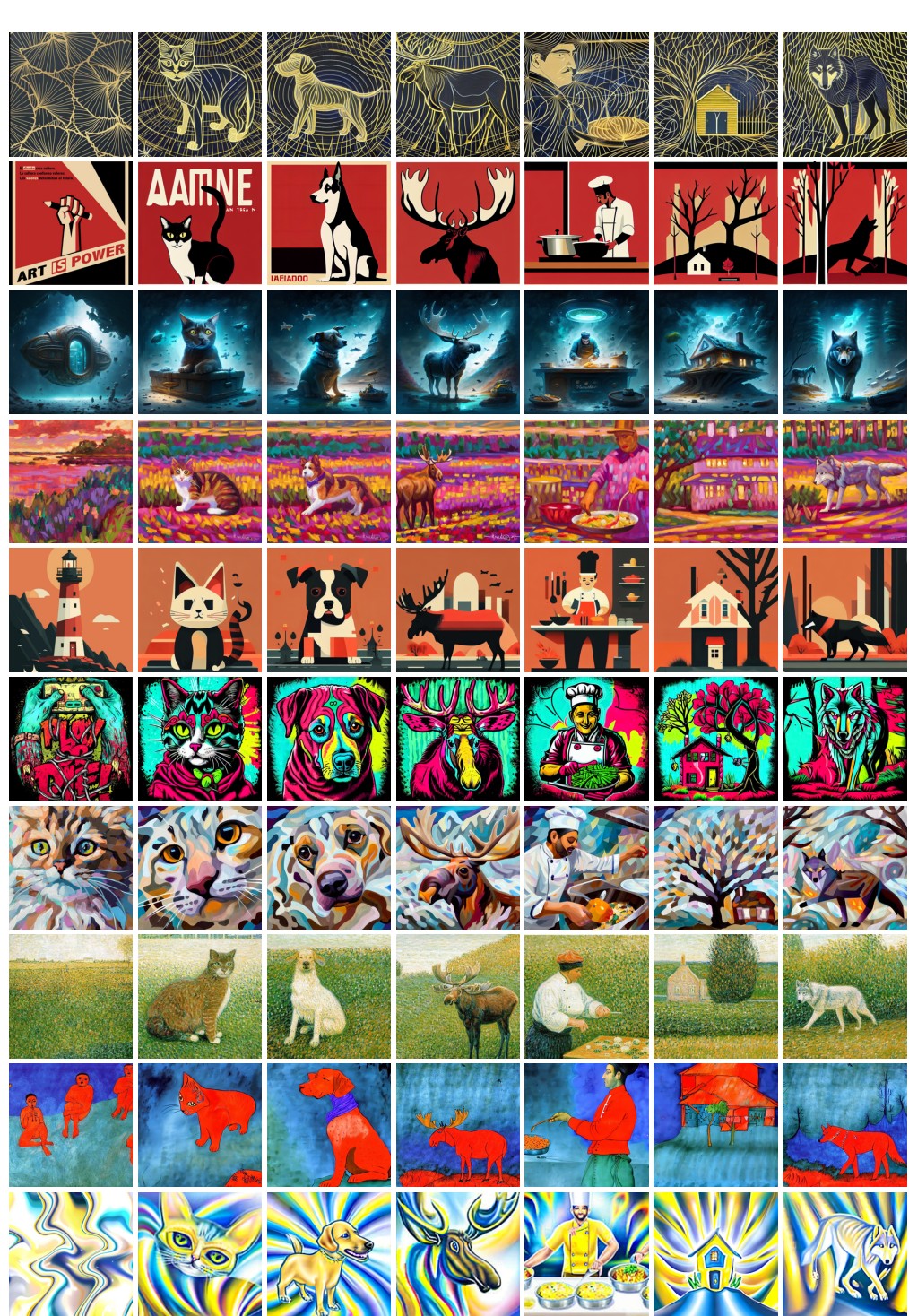

Figure 27: Additional text-driven style transfer visualization results of **StyleShot**. From left to right, Reference style image, "A cat", "A dog", "A moose", "A chef preparing meals in kitchen", "A house with a tree beside", "A wolf walking stealthily through the forest".

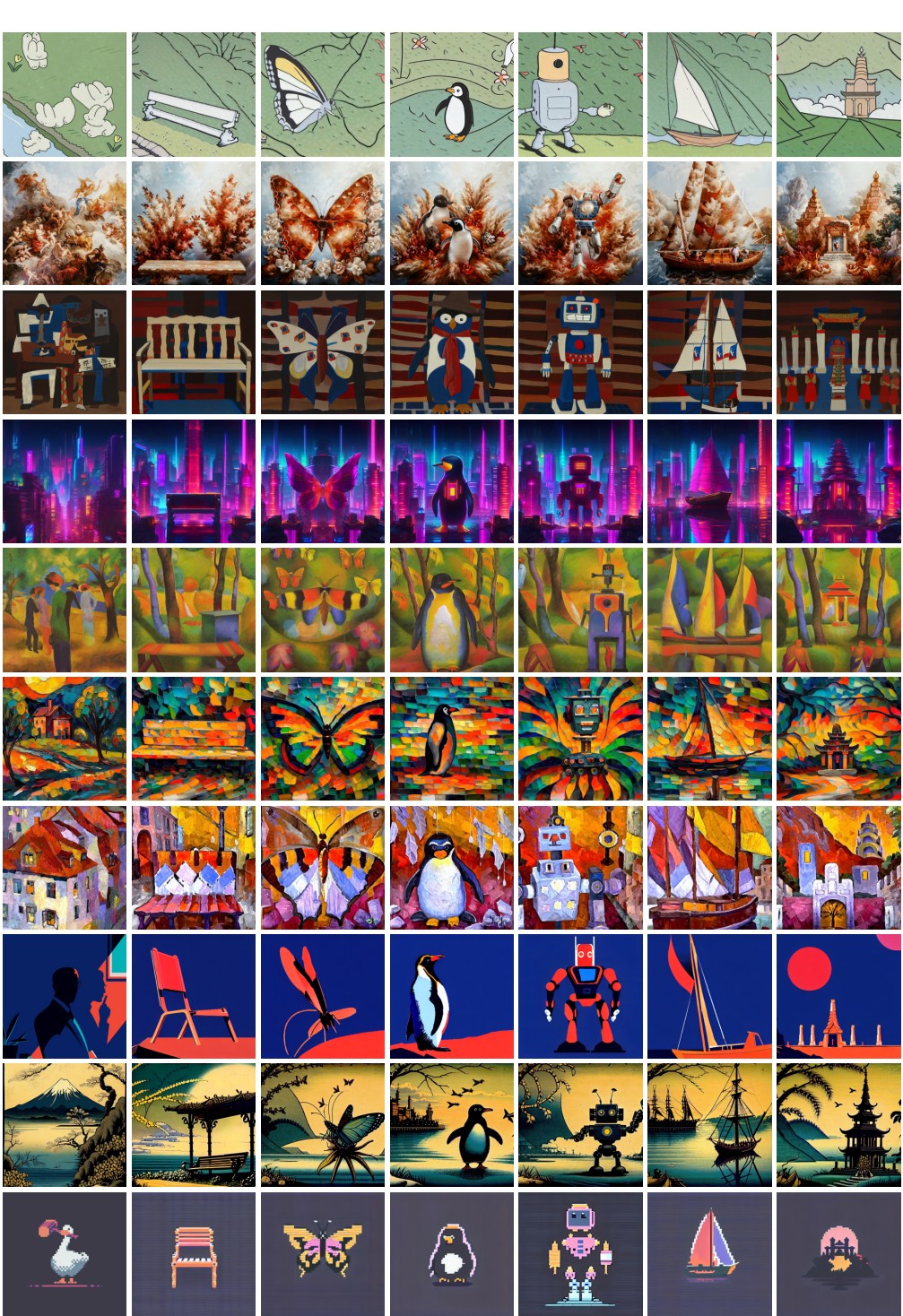

Figure 28: Additional text-driven style transfer visualization results of **StyleShot**. From left to right, Reference style image, "A bench", "A butterfly", "A penguin", "A robot", "A wooden sailboat docked in a harbor", "A ancient temple surrounded by lush vegetation".

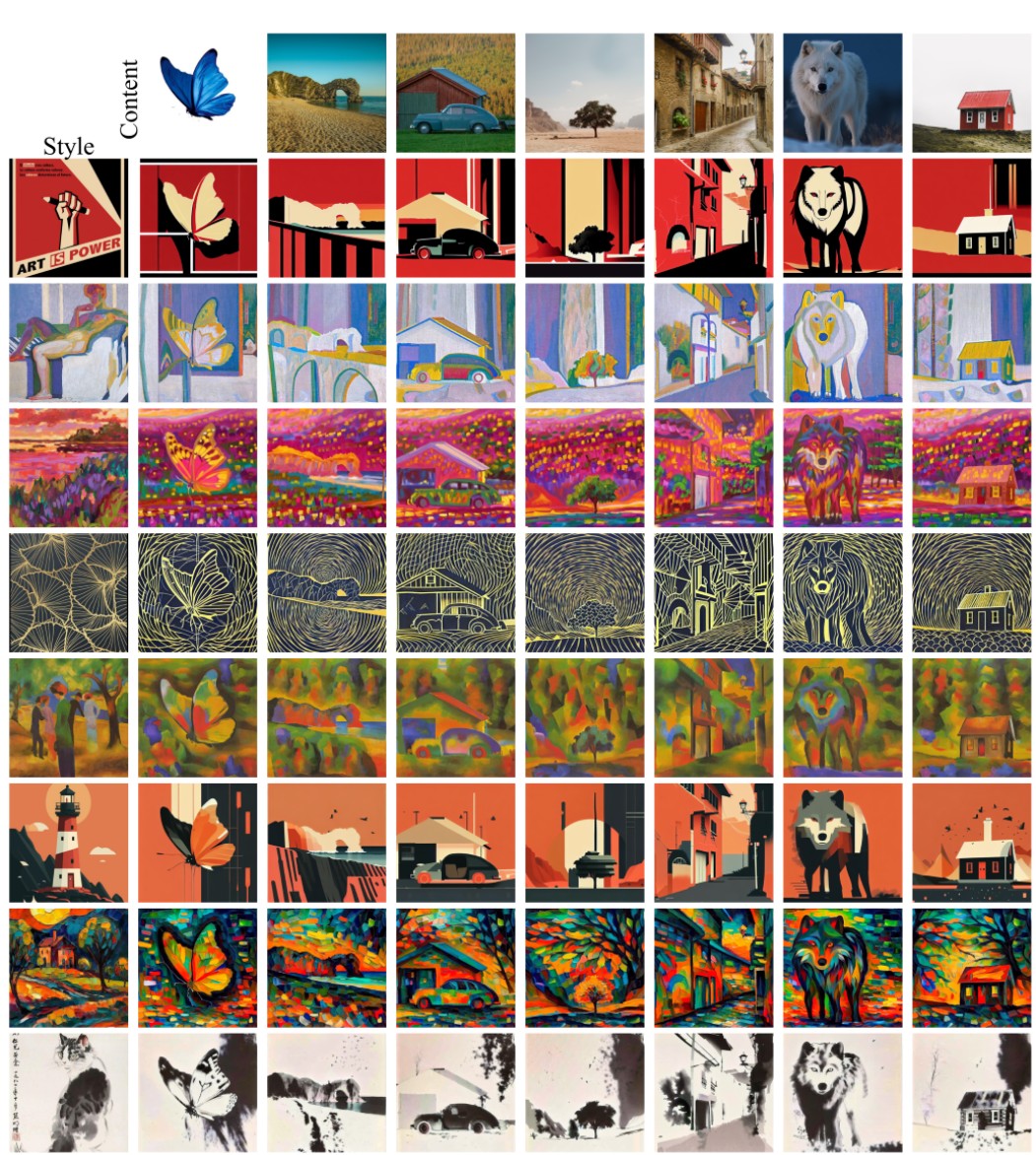

Figure 29: Additional image-driven style transfer visualization results of **StyleShot**.

