# OpenReview forum: "StyleShot: A snapshot on any style"
_ICLR.cc/2025/Conference — Submitted to ICLR 2025_

### Official Review · Reviewer_WDTr · 2024-11-02

**Soundness:** 2
**Presentation:** 3
**Contribution:** 2
**Rating:** 6
**Confidence:** 5

**Summary:**

This paper proposes a style transfer method that consists of: 1) using a pre-trained text-to-image model as the generation backbone, 2) employing multi-scale cropped patches from a style image as the style condition, and 3) using contour conditions from the content image to provide structural guidance. During the generation process, the model can also incorporate text descriptions as additional guidance. Additionally, the authors introduce a new style image dataset named StyleGallery, sourced from MidJourney and WIKIART. Comprehensive experiments are conducted to examine the effects of dataset selection and style image partition strategies. Empirical results suggest that the proposed method generates outputs that are more favorably received by human evaluators.

**Strengths:**

* The paper is easy to follow, and the qualitative results are well-presented.
* The authors introduce a new dataset, StyleGallery, which offers more diverse and balanced styles compared to the LAION-Aesthetic dataset. Notably, 99.7% of the samples in StyleGallery include style descriptions, making it a valuable resource for future style-related research. Additionally, the authors curated a subset of StyleGallery called StyleBench for the evaluation of style transfer, which could potentially serve as a standard benchmark.

**Weaknesses:**

1. The proposed StyleShot architecture does not show significant novelty. The overall structure primarily follows the established method of conditional diffusion generation by integrating style and content information into the foundational diffusion layers. This technique is widely used in diffusion-based transfer learning, such as ControlNet [A] and T2I-Adapter [B] (with this work aligning more closely with the latter).
2. Similarly, using patch-based encoding for the style image is not new. Previous works, such as [C, D], also employ transformer encoders for patch-wise style image encoding. While the improvement in this work lies in using multi-scale patch size, unlike prior works that use a single patch size. The reviewer believes this modification offers only limited novelty.
3. The authors emphasize their content extraction approach in the Method section. However, in the comparison shown in Figure 14, Canny, HED, and the proposed methods are evaluated using ControlNet, not the proposed content-fusion encoder. Therefore, the reviewer does not believe that Figure 14 adequately validates the proposed content extraction method as superior to Canny or HED, given that these are not compared under the same conditions as the proposed content extraction (as seen in the last column of Figure 14). Additionally, when Canny, HED, and the proposed method all use ControlNet as a condition (rows 3-5 of Figure 14), it is difficult to support the claim that the proposed method enables greater stylization (Ln 484-485).
4. Typo: Figure 14's caption, "Rows 3-5" --> "Columns 3-5." Also, it would be beneficial to provide more statistical details about the collected StyleGallery, such as the total number of images, total number of styles, and an analysis of the balance of styles within the dataset, etc.

[A] Adding conditional control to text-to-image diffusion models

[B] T2i-adapter: Learning adapters to dig out more controllable ability for text-to-image diffusion models

[C] StyTr 2 : Image Style Transfer with Transformers

[D] Master: Meta Style Transformer for Controllable Zero-Shot and Few-Shot Artistic Style Transfer

**Questions:**

My current rating for this paper is borderline accept. The reviewer acknowledges that the quality of the proposed style transfer is commendable, and the contribution of a new style dataset is a noteworthy addition. However, a higher rating is hindered by the limited methodological novelty (see weaknesses 1 and 2) and the lack of fairness in the experimental comparisons (see weakness 3).

---

> ### Author Response · Authors · 2024-11-22
> **Rebuttal by Authors**
>
> **W1: Overall structure does not show significant novelty.**
>
> As mentioned, conditional diffusion generation has already been used in certain style transfer tasks.
> However, the goal of StyleShot is to address the limitations of generalized style transfer based on a conditional diffusion model.
> We achieve this by designing a multi-scale style-aware encoder specifically for extracting rich and expressive features, while also highlighting the importance of a well-organized style-balanced dataset.
> Generally, we demonstrate a simple yet effective way to train a powerful generalized style transfer model without a lot of fancy modules. We believe these contributions are highly valuable to the community.
>
> **W2: Using patch-based encoding for the style image is not new**
>
> StyTR2 [1] and Master [2] achieve patch-based encoding with the help of the patch partitioning inherently implemented in transformer-like network without any specific design for style extraction. This implementation has many limitations, i.e., only one patch scale is used, which makes it difficult to perceive high-level styles; the network can access the full image while extracting style embedding since the content of this image is not broken, which makes the model tend to have content leakage issue under self-supervised learning.
>
> Instead, StyleShot adopts multi-scale partitioning. Each scale is encoded independently with partial access to the full image, which force the model to focus on the style itself instead of the semantic content. Multi-scale information compensate for each other for perceiving diverse image styles.
>
> Visual results in **Fig. 7, Fig. 8** confirm that StyleShot outperforms other baseline models, demonstrating the efficacy of our design in learning rich and expressive style representations.
> Furthermore, we evaluate Master (as Master is not open source, we use an unofficial GitHub implementation), as shown in **Fig. R2** and table below:
> | metrics | Master | StyleShot |
> | -------- | -------- | -------- |
> | clip image $\uparrow$     | 0.542     | **0.660**     |
> | Style loss $\downarrow$    | 9.824     | **7.872**     |
>
> StyleShot outperforms Master in both qualitative and quantitative comparisons.
>
> **W3: Experiments of content extraction**
>
> We train these three extractions with our proposed content-fusion encoder and present the visualizations in **Fig. R1**.
> The stylized results of Canny and HED preserve real-world textures due to their detailed edges.
> In contrast, the results generated from our Content Input are more closely aligned with the reference style.
>
> **W4: Typo & statistical details about StyleGallery**
>
> Thanks for your reminder. We have revised this typo in the new version.
>
> As described in our paper in **Sec. 3.4** and **Sec. B.1**, StyleGallery contains 5.7M text-image pairs and 99.7% of the images in our StyleGallery have style descriptions.
> After statistical analysis, we report that StyleGallery contains 0.351M identical style descriptions.
> For more details, these style descriptions can be categorized into three groups:
>
> Basic Styles: There are 372 basic styles, which include general or simple style definitions, such as basic textures, colors, or geometric features (e.g., "painting" and "photography"). Each basic style has more than 7k images.
>
> Advanced Styles: A total of 11,087 styles fall under this category. These styles may include intricate brushstrokes or specific artistic techniques (e.g., "watercolor painting" or "cinema photography"). Each advanced style has between 100 and 7k images.
>
> Personalized Styles: This category consists of 0.339M style descriptions. Personalized Styles are highly customized styles based on the specific needs or aesthetics of the user (e.g., "rich geometry" and "highly detailed pixel art"). Each personalized style has fewer than 100 images.
>
> Furthermore, the styles in each category follows a balanced distribution as shown in **Fig. 6** (right).These balanced and extensive styles in StyleGallery benefit our model in learning expressive and generalized style representations.
>
>
> [1] StyTr 2 : Image Style Transfer with Transformers
>
> [2] Master: Meta Style Transformer for Controllable Zero-Shot and Few-Shot Artistic Style Transfer

---

> > ### Comment · Reviewer_WDTr · 2024-11-24
> >
> > The reviewer acknowledges that the responses to W3 and W4 are satisfactory and address the concerns adequately.
> >
> > However, the responses for W1 and W2 remain insufficient to fully address my concerns:
> >
> > Regarding W1: The authors reiterate their contributions but fail to sufficiently refute the weakness highlighted in the review. Specifically, the concern remains that most of the network structure closely follows existing conditional generation diffusion pipelines. While listing contributions helps clarify the proposed method, it does not adequately address or overturn the reviewer's observation regarding the lack of significant structural novelty.
> >
> > Regarding W2: The reviewer acknowledges the improvement brought by the multiscale patch strategy, as already stated in the original review. However, the concern lies in the limited scope of this improvement. The authors' response primarily compares their method to one patch-based solution using CLIP image and style loss, but this comparison is insufficient for two reasons:
> >
> > 1. The baseline used involves a different network and training strategy (non-diffusion model), making direct numerical comparisons less compelling as justification.
> >
> > 2. While the ablation study partially validates the partition strategy, a more convincing experiment could involve comparing the proposed multiscale patch strategy against using a single-level patch under the same method. This would isolate the specific contribution of the multiscale approach within the context of the proposed diffusion framework.
> >
> > While the reviewer continues to acknowledge the utility of the multiscale patch strategy, its novelty and significance remain limited from my perspective.
> >
> > Given above concerns, the reviewer has decided to maintain my current rating.

---

> > > ### Author Response · Authors · 2024-11-25
> > >
> > > Dear Reviewer WDTr,
> > >
> > > Thank you for your feedback. We appreciate your positive comments on our rebuttal and will add the discussions to our revision. We are grateful for your guidance and support in helping us improve our work.
> > >
> > > However, we aim to clarify regarding W1  that existing diffusion-based style transfer methods (DEADiff-CVPR, StyleCrafter- SIGGRAPH Asia) are primarily designed within the framework of conditional diffusion models. These designs often fall short in achieving strong performance in generalized style transfer scenarios. To address this, StyleShot is proposed with the following key components:
> > >
> > > A style-aware encoder that captures rich and expressive style representations.
> > >
> > > A specially designed content input mechanism and a content-fusion encoder for better integration of content and style.
> > >
> > > A well-organized, style-balanced dataset.
> > >
> > > These components have been rarely explored in previous diffusion-based methods. Thus, it is unreasonable to critique a method solely because it represents an improvement within an established framework, especially given that the extensive results demonstrate its effectiveness.
> > >
> > > Regarding W2, we have demonstrated in Figure 13 that multi-scale patch designs outperform single-scale patches under the same model. The simplicity of the  multi-scale design does not diminish its contribution. Using these seemingly straightforward techniques, we have trained a robust and highly validated model. This is precisely one of the core messages we aim to convey in this paper.
> > >
> > > Best regards,
> > >
> > > The Authors

---

> > > > ### Comment · Reviewer_WDTr · 2024-11-25
> > > >
> > > > The reviewer thanks author's response, and my concern regarding on W2 is addressed. However, I am still not convinced enough for the contribution of the overall model design. My currenting would be slightly above the borderline accept but has not reached the accept line (**weak accept**, though the ICLR does not have this rating, the reviewer would like to request the AC to take this comment into consideration), considering the quantitative comparision for multi-scale patches has been justified, and together with the contribution of providing a new dataset.

---

> > > > > ### Author Response · Authors · 2024-11-26
> > > > >
> > > > > We are encouraged to hear that your concerns have been resolved.
> > > > > Thank you for your valuable feedback and for **recommending our paper for acceptance**. We sincerely appreciate your constructive suggestions and will revise the manuscript based on the discussion.

---

### Official Review · Reviewer_FjaZ · 2024-11-02

**Soundness:** 3
**Presentation:** 3
**Contribution:** 3
**Rating:** 6
**Confidence:** 4

**Summary:**

This paper proposes a generalized diffusion-based style transfer method called StyleShot which learns style from reference images and learns content from text prompts. The core component of StyleShot is a style-aware encoder which is designed to extract expressive style representation with decoupling training strategy. In addition, a new dataset StyleGallery is constructed to enhance the generalization ability of the proposed method.

**Strengths:**

1. The artistic images generated by the proposed method is impressive.
2. This paper provides a new style-balanced dataset.
3. The proposed method is simple yet effective.
4. Extensive experiments are conducted to evaluate the performance of the proposed method.

**Weaknesses:**

1. The quantitative results are not satisfying. The CLIP scores reported in Table 1 show that the proposed method is just comparable with previous methods (no gain). Although the authors claim that these metrics are not ideal for evaluation in style transfer tasks, they are still among the most widely used and authoritative metrics in this field.

2. When comparing with existing text-driven style transfer methods, several more state-of-the-art approaches (such as DreamStyler, T2I-Adapter, and InstantStyle) are only included in the supplementary material. Why not present them in the main paper? Moreover, some of these methods (such as DreamStyler), like the proposed method, can be used for both text-driven style transfer and image-driven style transfer, and thus a more comprehensive comparison with them on both tasks would be appropriate.

3. In the user study, how many users participated in the survey? How many samples were involved?

4. For the proposed de-stylization, this paper provides quantitative experiments in the supplementary material to demonstrate its effectiveness. However, it would be better to also include qualitative experiments to provide a more intuitive observation of its impact.

**Questions:**

Please see **Weaknesses**.

---

> ### Author Response · Authors · 2024-11-22
> **Rebuttal by Authors**
>
> **W1: The quantitative results are not satisfying.**
>
> The inadequacy of CLIP scores as evaluation metrics for style transfer is discussed in StyleDrop[1], **Section 4.2.1 paragraph 'CLIP scores'**, and StyleCrafter[2], **Section 4.1 paragraph 'Evaluation Metrics'**. Notably, simple image replication can yield a perfect CLIP image score, but image replication does not require any style transfer ability. This is the reason why CLIP score is not suitable for evaluating style transfer tasks. Therefore, to substantiate the efficacy of our method, we present extensive visual results and human feedback in both the main paper and appendices.
>
> [1] Styledrop: Text-to-image synthesis of any style, Sohn et al., NIPS 2024.
>
> [2] Stylecrafter: Enhancing stylized text-to-video generation with style adapter, Liu et al., arxiv 2023.
>
> **W2: Why not present other SOTA methods in main paper & image-driven style transfer of DreamStyler.**
>
> In this paper, we compared 11 text-driven SOTA style transfer methods. Including all of them in the main paper would make it overly crowded. For better visualization, results of some methods were moved to the Appendix, with explanations provided in the main paper.
> In **Fig. 7** (main paper) and **Fig. 20** (supp) StyleShot always achieves the best visual results, moving some methods to the Appendix **will not cause any misunderstanding of our paper**.
> However, we will consider Reviewer's suggestion and re-organized these 11 methods in the new version of our paper.
>
> Furthermore, we evaluate DreamStyler on image-driven style transfer in **Fig. R2** and table below:
> | metrics | DreamStyler | StyleShot |
> | -------- | -------- | -------- |
> | clip image $\uparrow$     | 0.578     | **0.660**     |
> | Style loss $\downarrow$    | 13.742     | **7.872**     |
>
> StyleShot outperforms DreamStyler in both qualitative and quantitative comparisons.
>
> **W3: Details of user study.**
>
> 22 users participated in our user study. As described in Appendix B.2, each user evaluated 60 samples across 30 tasks, resulting in a total of 1,320 samples.
>
> **W4: Qualitative experiments for de-stylization.**
>
> We provide the qualitative experiments for de-stylization in **Fig. R3**. The stylized results of StyleShot without de-stylization struggle to capture styles such as 3D or pixel art. This demonstrates that the style-related descriptions in the text hinder the model’s ability to learn style features from the reference image.

---

> > ### Comment · Reviewer_FjaZ · 2024-11-25
> >
> > I appreciate the rebuttals provided by the authors. I decided to keep my initial rating.

---

> > > ### Author Response · Authors · 2024-11-25
> > >
> > > Dear Reviewer FjaZ,
> > >
> > > Thank you for your feedback. We appreciate your positive comments on our rebuttal and will add the discussions to our revision. We are grateful for your guidance and support in helping us improve our work.
> > >
> > > Best regards,
> > >
> > > The Authors

---

### Official Review · Reviewer_MtZ9 · 2024-11-03

**Soundness:** 3
**Presentation:** 3
**Contribution:** 3
**Rating:** 5
**Confidence:** 5

**Summary:**

This manuscript describes a stylized image framework in an IP-adaptor fashion. Special attention is paid to learning style representation: a style-aware encoder is proposed. Based on Stable Diffusion, Styleshot uses a multi-scale patch partitioning scheme with MoE and ResBlocks to capture diverse style cues. It then processes these through Transformer Blocks and injects the style embeddings into the model. A style-balanced dataset, StyleGallery, is collected. Text- and image-driven stylization tasks are evaluated.

**Strengths:**

Overall, the reviewer tends to classify this paper's originality as removing limitations from prior results. The proposed method is reasonable and has good motivation. The focus of learning style representations for style transfer tasks is valid. Some of the generated results are impressive.

**Weaknesses:**

- Overall, the reviewer would not rate the technical contribution of the proposed method high. Based on the IP adaptor, the proposed method is somewhat incremental. It seems to be a successful attempt to combine MOE and IP adaptor.

- The reviewer is confused about the claim that the authors ''show that a good style representation is crucial'' for style transfer. Since CAST has already observed related conclusions and proved the usefulness of learning a good style encoder.

- Related to the first issue, the introduction is organized in a loose way. The authors are suggested to focus more on finding the position of the proposed method in the style transfer research context.

- Given that the proposed method is trained on the newly proposed dataset, the reviewer finds the comparison to be somewhat unfair.  Some results of the selected comparing methods are much worse than common sense, such as the 4th lines in figure 8. All other methods failed to learn the style. Please check and compare with more recent diffusion-based models such as Z-Star (CVPR 2024, also training free).

- NO user study of image-driven style transfer. The reviewer agrees with the authors that different methods were used to evaluate different unavailable datasets. How about considering using third-party dataset such as ''A Comprehensive Evaluation of Arbitrary Image Style Transfer Methods'', IEEE TVCG.

**Questions:**

The manuscript is presented in an easy-to-follow way. There are no more questions. I have just some suggestions for convincing the audience that the proposed method has good technical contributions; see weaknesses for details. The reviewer is open to valid responses and changing my initial ratings.

---

> ### Author Response · Authors · 2024-11-22
> **Rebuttal by Authors**
>
> **W1: Technical novelty**
>
> StyleShot incorporates an MOE-like structure for style extraction and parallel cross-attention proposed in IP-Adapter for style feature injection. However, they are not the focus of this paper. StyleShot aims to tell a good style representation is important for *generalized style transfer*, and especially elaborate how to establish powerful style representation for open-domain images through our multi-scale patching strategy and style encoder. Moreover, StyleShot points out the significance of building a style-balanced dataset for style transfer tasks, and demonstrate the importance with thorough experimental analysis.
>
> Overall, StyleShot shows a simple yet effective way to train *a powerful generalized style transfer model*, which beats all existing training-free and training-based methods, with our designated modules instead of a lot of fancy techniques. We believe all of these constitute solid contribution for the community.
>
> **W2: Style encoder in CAST**
>
> StyleShot and CAST differ in many aspects.
> - Our claim is that "a good style representation is crucial and sufficient for **generalized style transfer**" while CAST concludes that "a suitable style representation is essential to achieve satisfactory results". Though similar, we emphasize the style representation for generalized style transfer.
>
> - The style encoder of CAST is trained on a 30-class dataset (most of which are paintings). Although CAST[1] demonstrates the usefulness of training a good style encoder on these 30 categories, the performance on generalized style transfer is in question as CAST does not give any verification. On the contrary, we assess our model on a variety of styles.
>
> - Third, the style encoder of CAST adopts supervised training paradigm, improving the generalization ability by collecting more diverse labeled style data, which is extremely costly. In contrast, StyleShot learns styles from images in a self-supervised manner, effectively overcoming the limitations of the dataset, and can be easily scaled up.
>
> - Fourth, beyond the above, we also propose a well-organized style-balanced dataset and an effective multi-scale style-aware encoder for extracting rich and expressive style representations, with comprehensive experimental support.
>
> - Finally, our evaluation demonstrates that CAST performs poorly on styles outside of paintings as shown in the **Fig. 8**, while StyleShot achieves superior performance compared to other SOTA methods on a wide range of styles.
>
> **W3: The organization of introduction**
>
> Thank you for your suggestions. We will consider all reviewers' feedback and refine the introduction accordingly.
>
> **W4: Fair comparison & Z-Star**
>
> For fair comparison, we re-trained  StyleCrafter on StyleGallery, while other diffusion-based baselines either require test-time style-tuning or are not open source.
> As shown in **Fig. 10**, StyleShot achieves superior performance compared to StyleCrafter.
> We also observe that StyleCrafter (scale=1.0) has a content leakage issue.
>
> We have carefully checked that all results strictly follow the inference script of each method's official GitHub repo.
> Moreover, we observe that these methods perform well on painting styles, as shown in **Fig. R4** and **Fig. R5**.
> However, they perform poorly on high-level styles, as shown in **Fig. 7, Fig. 8**.
> This demonstrates that these poor results are due to their limited generalization ability in handling high-level styles, rather than issues with our implementations.
> In contrast, due to our style-aware encoder which learns the rich and expressive style representations, StyleShot is well generalized to the wide range of styles.
>
> We also present the results of Z-Star in **Fig. R2** and table below:
> | metrics | Z-Star | StyleShot |
> | -------- | -------- | -------- |
> | clip image $\uparrow$     | 0.607     | **0.660**     |
> | Style loss $\downarrow$    | 11.223     | **7.872**     |
>
> StyleShot outperforms Z-Star in both qualitative and quantitative comparisons.

---

> ### Author Response · Authors · 2024-11-22
> **Rebuttal by Authors**
>
> **W5: User study & evaluation on third-party bench**
>
> The user study of image-driven style transfer is presented in table below:
>
> | metric | AdaAttN | EFDM |StyTR-2 | CAST | InST | StyleID | StyleShot |
> | -------- | -------- | -------- | -------- | -------- | -------- | -------- | -------- |
> | human style $\uparrow$  | 0.075     | 0.112     | 0.129     | 0.135     | 0.093     | 0.040     | **0.416**     |
>
> StyleShot achieves the highest style alignment score.
>
> The third-party bench in [2] is a painting-dominant benchmark biased to ten painting styles such as impressionism and expressionism, while our StyleBench covers 73 distinct styles, ranging from paintings, flat illustrations, 3D rendering to sculptures with varying materials.
>
> As shown in **Fig. R4**, **Fig. R5**, StyleShot achieves the stable and superior performance on this third-party bench.
> We also provide the quantitative results in the tables below for reference only:
>
> | metric | AdaAttN | EFDM |StyTR-2 | CAST | InST | StyleID | StyleShot |
> | -------- | -------- | -------- | -------- | -------- | -------- | -------- | -------- |
> | clip image $\uparrow$  | 0.470     | 0.428     | 0.464     | 0.539     | 0.490     | 0.551     | **0.603**     |
> | style loss $\downarrow$  | 13.140     | 14.832     | **12.749**     | 13.117     | 14.122     | 14.881     | 15.415     |
>
> | metric | DEADiff | DeamBooth |InST | Style-Aligned | Style Crafter | StyleDrop | StyleShot |
> | -------- | -------- | -------- | -------- | -------- | -------- | -------- | -------- |
> | clip image $\uparrow$  | 0.552     | **0.761**    | 0.606     | 0.698     | 0.721     | 0.558    | 0.602     |
> | clip text $\uparrow$  | **0.232**     | 0.171     | 0.207     | 0.204     | 0.189     | 0.232     | 0.210    |
> | style loss $\downarrow$  | 24.237     | 7.668     | 14.570     | 6.330     | **4.056**     | 5.279     | 4.412     |
>
> [1] Domain Enhanced Arbitrary Image Style Transfer via Contrastive Learning, ACM SIGGRAPH
>
> [2] A Comprehensive Evaluation of Arbitrary Image Style Transfer Methods, IEEE TVCG.

---

> ### Author Response · Authors · 2024-12-01
> **Please let us know if your concerns have been addressed**
>
> Dear Reviewer MtZ9,
>
> Thank you again for your review. **We hope that our rebuttal could address your questions and concerns**, such as 'Technical novelty', 'Style encoder in CAST', 'The organization of introduction', 'Fair comparison & Z-Star'and 'User study & evaluation on third-party bench'. As the discussion phase is nearing its end, **we would be grateful to hear your feedback and wondered if you might still have any concerns we could address**.
>
> **It would be appreciated if you could raise your score on our paper if we address your concerns**. We thank you again for your effort in reviewing our paper.
>
> Best regards,
>
> The Authors

---

### Official Review · Reviewer_KuS9 · 2024-11-04

**Soundness:** 3
**Presentation:** 3
**Contribution:** 3
**Rating:** 5
**Confidence:** 4

**Summary:**

This paper presents an encoder to extract style info from the reference style. The information is combined with text embedding and injects to diffusion models through the cross-attention layer. Meanwhile, to preserve content, contours are extracted and inject to the diffusion network through controlnet-like residue addition. Experimental results were compared with SOTA methods such as InstantStyle, StyleAligned, etc.

**Strengths:**

The results look interesting and the method is straightforward.

**Weaknesses:**

In Figure 7, the results from StyleAligned look weird, can you detail how the results were generated?
While training for the style encoder, it seems that a crafted datasets were used. Can you detailed the model training details including the encoder as well as the feature injection part in the diffusion model?

**Questions:**

Are text-based stylization results generated using the same model and set the content reference as black or is this a different model?

---

> ### Author Response · Authors · 2024-11-22
> **Rebuttal by Authors**
>
> **W1: Details of Style-Aligned**
>
> The stylized results of Style-Aligned strictly follow its inference script. Issue #27 in Style-Aligned's official GitHub repo also reports the weird results. Furthermore, following reviewer MtZ9's suggestion, we conducted the evaluation on the third-party dataset, which primarily consists of painting styles. As shown in **Fig. R4**, Style-Aligned performs well on painting style. However, it generates weird results for other high-level styles in **Fig. 7**.
> Different from that, due to our style-aware encoder which learns the rich and expressive style representations, StyleShot is well generalized to the wide range of styles.
>
> **W2: Details of training**
>
> We have already explained implementation details in **Sec. 3** in the main paper and **Sec B.1** in Appendix.
> Specifically, our model is trained in two stages. In the first stage, we train only the style-aware encoder and the cross-attention module for style injection. In the second stage, we freeze the style-aware encoder and the corresponding style injection module, and train the content-fusion encoder. We will release the code in the future. If you have any other specific questions about model training, please feel free to raise them in the discussion.
>
> **Q1: Details of text-based stylization**
>
> Text-based stylization is generated from the same model and disable the content input by setting the content reference as black and content scale to zero.

---

> ### Author Response · Authors · 2024-11-25
> **Please let us know if your concerns have been addressed**
>
> Dear Reviewer KuS9,
>
> Thank you again for your review. **We hope that our rebuttal could address your questions and concerns**, such as 'Details of Style-Aligned', 'Details of training' and 'Details of text-based stylization'. As the discussion phase is nearing its end, **we would be grateful to hear your feedback and wondered if you might still have any concerns we could address**.
>
> **It would be appreciated if you could raise your score on our paper if we address your concerns**. We thank you again for your effort in reviewing our paper.
>
> Best regards,
>
> The Authors

---

> > ### Comment · Reviewer_KuS9 · 2024-11-27
> > **Re: Please let us know if your concerns have been addressed**
> >
> > Thanks for your response! I decided to keep my initial rating. Thanks.

---

> > > ### Author Response · Authors · 2024-11-27
> > > **Seeking further discussion**
> > >
> > > Dear Reviewer KuS9,
> > >
> > > Thanks for your response!
> > >
> > > Could you kindly let us know **which issue was not sufficiently addressed in our rebuttal**. We would like to respectfully remind you that a rating of 5 indicates *marginally below the acceptance threshold*.
> > > If you have any additional concerns, we would be more than happy to discuss them further. As ICLR is an open platform for thorough academic exchange, we trust that all reviewers are committed to helping authors improve the quality of their manuscripts. We are eager to understand the reasons behind your decision to maintain the rating. **We will be attentively awaiting your reply until the rebuttal deadline on December 2nd**.
> > >
> > > Best regards,
> > >
> > > The Authors

---

### Author Response · Authors · 2024-11-22
**General comments**

Thank all reviewers for reviewing and providing constructive feedback to our paper. We appreciate your acknowledgment of our method simple yet effective (Reviewer KuS9, FjaZ), reasonable and well motivated (Reviewer MtZ9), our proposed style-balanced dataset meaningful and helpful (Reviewer FjaZ, WDTr), which leads to an impressive and satisfactory style transfer visual results (Reviewer KuS9, MtZ9, FjaZ, WDTr).

Please note that, we put the rebuttal pdf both in the Appendix of revised manuscript and the supplemental materials. In the rebuttal pdf, we included five figures:

* Figure R1: Visual results for different content extractions on ControlNet or our proposed content-fusion encoder.
* Figure R2: Visual comparison among StyleShot, DreamStyler, Z-Star and Master.
* Figure R3: Visual results with or without de-stylization.
* Figure R4: Qualitative comparison with SOTA text-driven style transfer methods on third-party bench from [1].
* Figure R5: Qualitative comparison with SOTA image-driven style transfer methods on third-party bench from [1].

We respond to each reviewer below to address the concerns. Please take a look and let us know if further clarification / discussion is needed.

Also, we will include all these discussions in the next version and release codes, models and training data of StyleShot.

[1]. ''A Comprehensive Evaluation of Arbitrary Image Style Transfer Methods‘’, IEEE TVCG.

---

### Meta-Review · Area_Chair_b2CW · 2024-12-20

**Metareview:**

The reviewers share concerns regarding the modest technical contributions and novelty, and the insufficient experimental comparisons and performance evaluations presented in the paper. Although the authors addressed some of the issues during the rebuttal, the crucial concerns about the technical contributions and the insufficient experimental comparisons are not fully addressed. As a result, the paper does not meet the acceptance threshold at this time. The authors are encouraged to carefully consider the reviewers' feedback when revising the paper.

**Additional Comments On Reviewer Discussion:**

The rebuttal addressed some of the reviewers' concerns. However, the crucial concerns about its technique contribution and insufficient experimental comparisons are not fully addressed.

---

### Decision · Program_Chairs · 2025-01-22

Reject